# Metabolic modelling reveals increased autonomy and antagonism in type 2 diabetic gut microbiota

A Samer Kadibalban [1,6], Axel Künstner [2], Torsten Schröder [3,4], Julius Zauleck [3], Oliver Witt [3], Georgios Marinos [1,5] & Christoph Kaleta [1✉]

## Abstract

**Type 2 diabetes (T2D) presents a global health concern, with evidence highlighting the role of the human gut microbiome in metabolic diseases. This study employs metabolic modelling to elucidate changes in host–microbiome interactions in T2D. Glucose levels, diet, 16S sequences and metadata were collected for 1866 individuals. In addition, microbial community models, and ecological interactions were simulated for the gut microbiomes. Our findings revealed a significant decrease in metabolic fluxes provided by the host's diet to the microbiome in T2D patients, accompanied by increased within-community exchanges. Moreover, the diabetic microbiomes shift towards increased exploitative ecological interactions at the expense of collaborative interactions. The reduced microbiome-to-host butyrate flux, along with decreased fluxes of amino acids (including tryptophan), nucleotides, and B vitamins from the host's diet, further highlight the dysregulation in microbial-host interactions in diabetes. In addition, microbiomes of T2D patients exhibit enrichment in energy metabolism, indicative of increased metabolic activity and antagonism. This study sheds light on the increased microbiome autonomy and antagonism accompanying diabetes, and provides candidate metabolic targets for intervention studies and experimental validation.**

**Keywords** Diabetes; T2D; Microbiome; Metabolic Modelling; Gut Microbial Community
**Subject Categories** Computational Biology; Metabolism; Microbiology, Virology & Host Pathogen Interaction

## Introduction

Despite the increased awareness as well as the development and availability of diagnostic and treatment approaches, type 2 diabetes (T2D) still presents an increasing global health issue. It was estimated that in 2021, around 537 million adults were living with diabetes, and this is forecasted to reach 783 million by 2045 (Xie et al, 2022). The global increase in T2D is facilitated by the continuous transition into an industrialised lifestyle that intrinsically affects people's daily activities and dietary habits. The human gut is one of the densest known microbial habitats. With a high biodiversity and a genetic repertoire of over three million genes and thousands of metabolites (Rinninella et al, 2019), this ecosystem is characterised by a complex network of metabolic interactions between the microbial community and the host. This crosstalk shapes the relations among the microbial species and plays a crucial role in the host's health.

Moreover, the gut microbial community profoundly affects human metabolism. Changes in the gut microbiota are associated with altered glucose regulation, which is closely linked to the development of type 2 diabetes and its related complications (Cunningham et al, 2021). In addition, the impact of the gut microbiota extends to various physiological processes, including chronic low-grade inflammation, β-cell dysfunction, gut permeability, glycolipid metabolism, and insulin sensitivity, all of which are pertinent to T2D development (Pan et al, 2021). The mutual influence between T2D and the gut microbial community is bidirectional, as elevated blood sugar levels can heighten intestinal permeability via a glucose transporter, altering the gut microbiome and triggering an inflammatory response (Ghosh et al, 2020). Moreover, there's a growing body of evidence suggesting that it's not only the microorganisms themselves but also their metabolites that contribute to the development of metabolic syndrome and diabetes (Herrema and Niess, 2020). Hence, understanding the complex relationship between the gut microbial community and diabetes could potentially lead to new strategies for prevention and treatment. However, while certain bacterial genera, such as *Ruminococcus*, *Fusobacterium*, *Blautia* and *Clostridium* have been reported in higher abundances among patients with T2D (Cunningham et al, 2021; Gurung et al, 2020), more research is needed to fully understand the complex interactions and interplay between pathomechanisms of T2D and the microbial community. Moreover, the influence of host-derived metabolites on the gut microbial community still needs to be more thoroughly investigated (Michellod and Liebeke, 2024).

Genome-scale metabolic models are powerful tools that provide comprehensive representations of the entire metabolic networks of

[1]Medical Systems Biology Group, University Hospital Schleswig-Holstein Campus Kiel & Kiel University, Kiel, Germany. [2]Medical Systems Biology Group, Lübeck Institute for Experimental Dermatology, University of Lübeck, Lübeck, Germany. [3]Perfood GmbH, Lübeck, Schleswig-Holstein, Germany. [4]Institute of Nutritional Medicine, University Hospital Schleswig-Holstein, Lübeck Campus & University of Lübeck, Lübeck, Germany. [5]CAU Innovation GmbH, Kiel University, Kiel, Germany. [6]Present address: Institute of Clinical Chemistry, University Hospital Schleswig-Holstein, Kiel Campus, Kiel, Germany. ✉E-mail: c.kaleta@iem.uni-kiel.de

organisms. Those models are reconstructed by associating the organism's genomic content with enzymatic reactions and metabolic transport (Zimmermann et al, 2021). The use of metabolic modelling through flux balance analysis is being increasingly employed to investigate complex metabolic networks and to study the metabolic interactions within microbial communities and with the host, especially in complex diseases such as Parkinson's and Alzheimer's (Zacharias et al, 2022; Forero-Rodríguez et al, 2022) as well as inflammatory bowel disease (IBD) (Aden et al, 2019, 2017). Nonetheless, the use of metabolic modelling to understand the alterations in the metabolic exchange profile between the gut microbiome and the human host in the context of diabetes is still limited to a few attempts (Ezzamouri et al, 2023; Väremo et al, 2013; Proffitt et al, 2022; Beura et al, 2024). Using metabolic modelling, reduced metabolic flux of acetate, butyrate, vitamin B5, and bicarbonate were reported in microbial community models of patients with type 2 diabetes, and those are presumed to have an impact on carbohydrate metabolism (Beura et al, 2024). Arguably, glucose stands out as the most crucial energy carrier and carbon source, serving as a foundation for metabolites and the construction of biopolymers across all kingdoms of life (Jeckelmann and Erni, 2020).

In this study, differential species abundances in the gut microbiome of a cohort of diabetic and healthy individuals were investigated. Thereafter, metabolic modelling was employed to predict the ecological interactions among the gut microbiomes as well as the microbial community metabolic fluxes. Subsequently, the microbiome metabolites associated with blood glucose measures and the production of those metabolites by dominant bacterial genera were observed. Finally, based on the observed increased glucose levels in the gut environment accompanying diabetes (Takeuchi et al, 2023), we hypothesise that the high availability of glucose as a simple energy source for the bacteria in the gut alters the microbial ecological interactions and metabolic exchanges. It was observed in this work that those alterations include an increase in antagonism, energy metabolism and autonomy of the gut microbiome, which becomes less shaped by the host's diet and more dependent on metabolic exchanges among its species.

## Results

### Associations of individual species with diabetes and no differences in diversity measures

The comparison of bacterial abundance between diabetic and healthy individuals revealed three species from distinct genera (*Clostridium celatum*, *Eubacterium rectale* and a *Lachnospiraceae* species) exhibiting significantly higher abundances in healthy individuals (Mann–Whitney *U* test *P* values are, respectively: $1.96 \times 10^{-5}$, $3.19 \times 10^{-2}$, and $2.01 \times 10^{-2}$). Conversely, diabetic patients showed higher abundances for eight species across five genera, as determined by Mann–Whitney *U* test, including *Ruminococcus gnavus* (*P* value = $6.34 \times 10^{-6}$), *Ruminococcus torques* (*P* value = $4.73 \times 10^{-3}$), *Granulicatella adiacens* (*P* value = $7.87 \times 10^{-6}$), *Clostridium clostridioforme* (*P* value = $4.57 \times 10^{-5}$), *Bacteroides vulgatus* (*P* value = $2.25 \times 10^{-2}$) and three species of *Streptococcus* (*Streptococcus australis*, *P* value = $8.94 \times 10^{-4}$, *Streptococcus mitis*, *P* value = $1.01 \times 10^{-3}$ and *Streptococcus parasanguinis*, *P* value = $1.28 \times 10^{-2}$) (Fig. 1A). Linear models that accounted for confounders (gender, age, waist-to-hip ratio,

activity, and antidiabetic medication treatment) did not yield any significant association between blood glucose and alpha diversity indexes (Shannon index *P* value = 0.08, Chao1 *P* value = 0.08, Species richness *P* value = 0.32, Simpson's *P* value = 0.09). Although there is a significant increase in the alpha diversity measures in diabetic patients compared to healthy individuals, these differences disappear when correcting for the confounders using linear models, with the exception of the species richness, which was significantly increased in diabetic individuals (Shannon index *P* value = 0.051, Chao1 *P* value = 0.14, Species richness *P* value = 0.003, Simpson's *P* value = 0.5). Moreover, no significant differences were observed between patients who are taking diabetes medications and those who are not using medication (Appendix Table S1). Beta diversity, on the other hand, did not show any clear signal of differences between diabetic and healthy individuals.

### A relative increase in exploitative ecological interactions corresponding to glucose levels

Upon application of the EcoGS pipeline to the microbial communities, ecological relations were predicted for each pair of species within each community. These predictions were weighted by the abundance of the tested pair. Subsequently, the frequency of each type of ecological interaction within each community was estimated. Amensalistic ecological interactions accounted for the highest percentage of all ecological interactions, followed by antagonism, competition, commensalism, mutualism and neutralism, respectively (Fig. 1B). By utilising linear models while controlling for confounders, a shift towards exploitative ecological interactions (antagonism and competition) at the expense of collaborative ecological interactions (mutualism and commensalism) was detected in association with increased glucose levels. The associations between ecological interaction ratios and glucose levels had the following *P* values: Antagonism/Commensalism (*P* value = 0.0097), Antagonism/Mutualism (*P* value = 0.0097), Competition/Commensalism (*P* value = 0.016) and Competition/Mutualism (*P* value = 0.016) (Fig. 1C). Furthermore, a shift towards amensalism, antagonism, and competition (exploitative interactions) at the expense of mutualism (collaborative) was observed in diabetic individuals, along with an increase in antagonism (exploitative) relative to commensalism (collaborative). Meanwhile, in healthy individuals, amensalism was increased relative to competition and antagonism, as determined by the Mann–Whitney *U* test (Fig. 1D; Appendix Fig. S1).

### Microbial metabolic fluxes are associated with blood glucose levels and diabetes

For each microbial community, a maximum of 243 within-community fluxes and 243 metabolic fluxes between the community and the host were predicted using community flux balance analysis (Pryor et al, 2019). The predicted fluxes with the host are either from the host's diet to the community (diet-to-community fluxes) or from the community to the host (community-to-host fluxes). Missing exchange reactions in certain communities were given a flux value of 0. After filtering fluxes based on standard deviation and presence in diabetic samples, 105 within-community, 43 community-to-host, and 59 diet-to-community fluxes were tested for their association with host blood glucose levels.

Linear regression analysis was used to assess the relationship between metabolic fluxes and blood glucose levels, considering confounding variables, namely gender, age, waist-to-hip ratio,

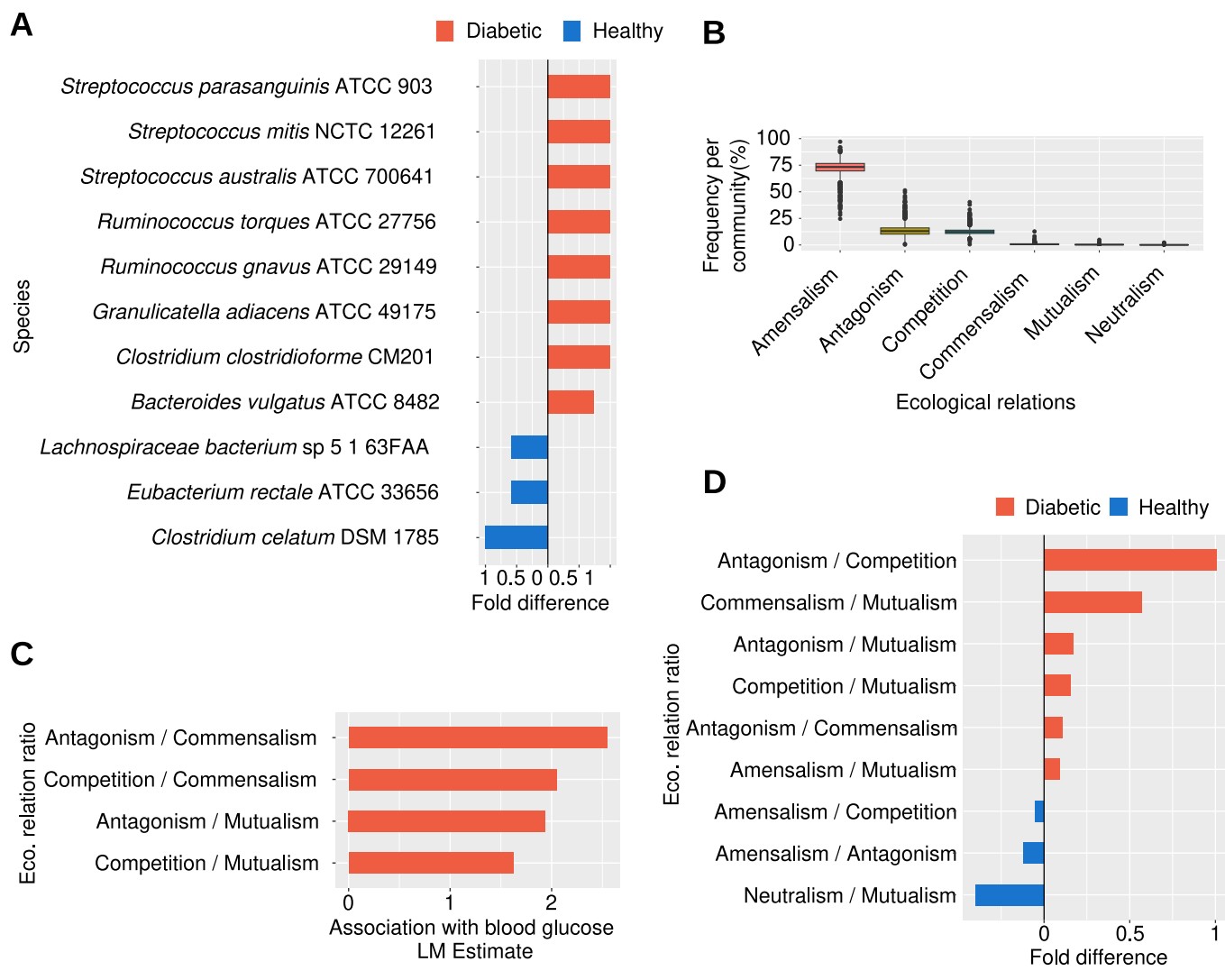

**Figure 1. Microbial diversity and ecological interactions vs blood glucose and diabetes.**

(A) Bacterial species with abundances that are significantly different between healthy and diabetic individuals, the fold difference was calculated as a log2 scale (Appendix Table S3). (B) Distribution of the frequencies for each of the six ecological interactions per microbial community. The black line in the middle of each box represents the median value, the bottom and top box edges represent the first and third quartals, respectively, the dots above and below the boxes represent potential outliers and the last points at the bottom and at the top of each box represent the minimum and maximum values, respectively. The number of samples per box plot is equal to the number of microbial communities in the study ($n = 2117$). (C) Ecological interaction ratios that are significantly associated with blood glucose (Appendix Table S3). The estimate of the linear model is shown on the X axis. (D) Ecological interaction ratios that are significantly different between diabetic and healthy individuals (Appendix Table S3). The fold difference of the ratios is on the X axis in a log2 scale. Source data are available online for this figure.

physical activity and diabetic medication. Following multiple testing corrections (false discovery rate control, FDR), 71 metabolic fluxes were significantly different between diabetic patients and healthy individuals, using the Mann–Whitney $U$ test with $\alpha = 0.05$ (Appendix Fig. S2). Those metabolites were grouped into 15 metabolic groups (Fig. 2A; Appendix Table S2). In addition, 31 metabolic fluxes were significantly associated with blood glucose levels (Fig. 2B; Appendix Table S3).

Investigating the fluxes of metabolites that are consumed by the community revealed that only four diet-to-community fluxes exhibited positive associations with blood glucose levels, namely D-glucose ($P$ value = 0.0087), glutamine ($P$ value = 0.045), lactate ($P$ value = 0.03) and fructose ($P$ value = 0.0307). Conversely, the remaining eleven diet-to-community fluxes associated with blood glucose had negative associations. On the contrary, all of the eleven within-community fluxes associated with blood glucose had a positive association (Appendix Table S3). Remarkably, among those metabolites, three have decreased diet-to-community fluxes while simultaneously increasing in their within-community fluxes accompanied with higher blood glucose. Those metabolites are: Spermidine, which has a role in alleviating oxidative stress and biofilm formation (Thongbhubate et al, 2021; Kumar et al, 2022), folate which is necessary for nucleic acid biosynthesis, amino acid homoeostasis, and methylation (Mölzer et al, 2021), Kok et al, 2020) and 3-deoxyadenosine which is essential for energy metabolism, replication, and transcription.

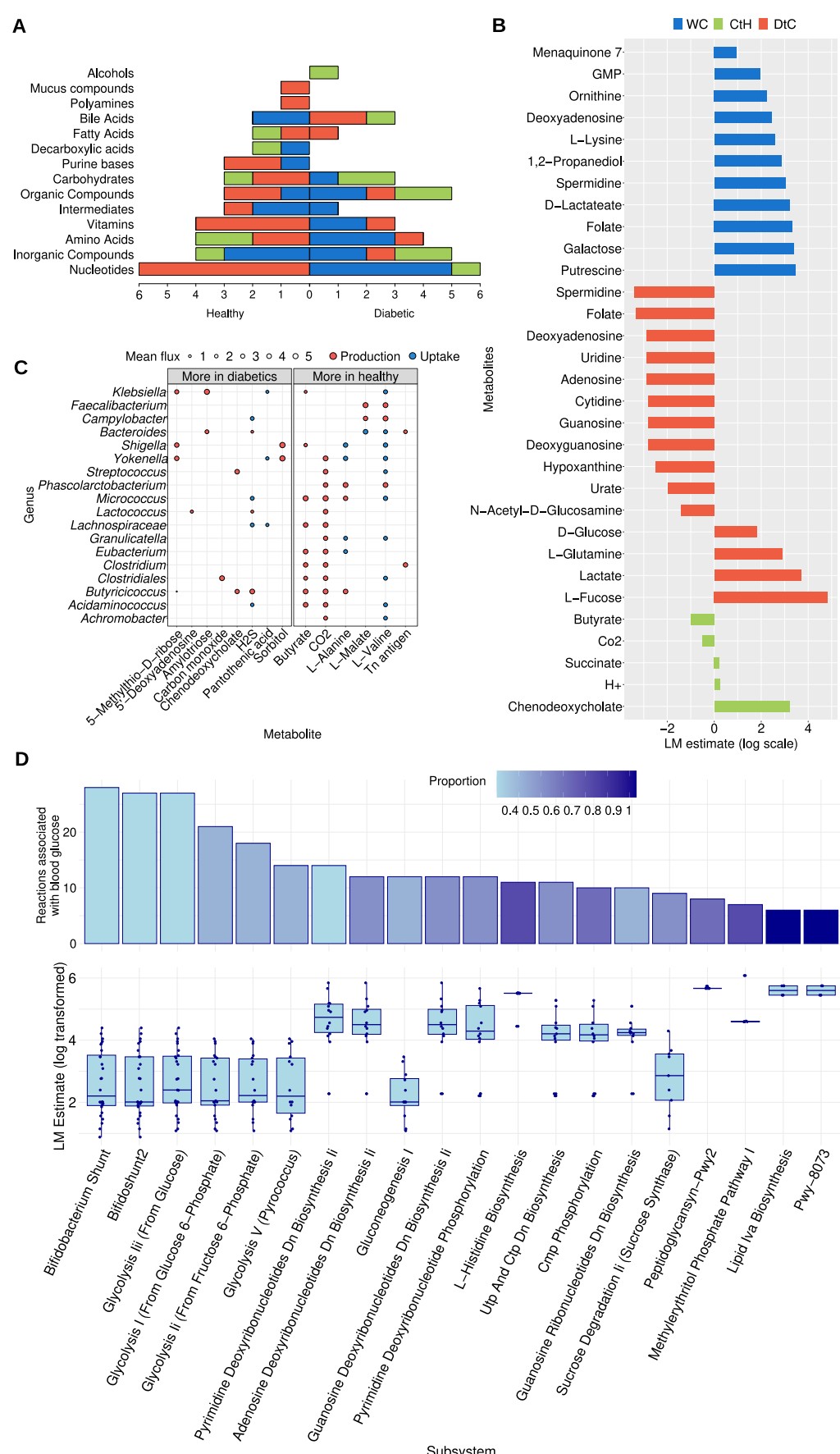

**Figure 2. The simulated metabolic exchange fluxes vs. blood glucose levels and diabetes.**

(A) The number of metabolic fluxes that are significantly different between diabetic and healthy individuals in each metabolic group (Appendix Table S2), the actual metabolites are shown in Appendix Fig. S3 and Appendix Table S3. The type of metabolic exchange (within-community, community-to-host, and diet-to-community) are represented respectively in blue, green and red as stacked bars. Bars pointing to the right represent metabolites increased in healthy individuals, and those pointing to the left are increased in diabetic patients. (B) The microbial metabolic fluxes that are significantly associated with the host blood glucose. Bars of negative associations point to the left, and bars of positive associations point to the right. The colours represent the three types of exchanges as in the previous plot A (WC within-community, CtH community-to-host, DtC diet-to-community). The X axis represents the estimated value of the linear model (coefficient of correlation). For visualisation purposes, those estimates were scaled as follows: log 10(|estimate| × sign(estimate)). (C) Microbial production and uptake of metabolites that are associated with blood glucose levels. The average metabolic flux is on the X axis, and the selected genera are on the Y axis. The size of the circles represents the flux rate (log10 scale), with red dots for metabolites produced by the bacteria and blue dots for metabolites taken up by the bacteria. The left side shows metabolic fluxes that are higher in diabetic communities, and the right shows metabolic fluxes that are higher in healthy individuals. (D) Enriched microbial metabolic subsystems. The upper bar plots represent the number of reactions associated with blood glucose in each subsystem, and the colour gradient of the bars reflects the proportion of the subsystem metabolites that are associated with blood glucose. The box plots in the lower part show the LM estimates for the reaction associations to blood glucose in each subsystem. The black lines in the middle of each box plot represent the median value, the bottom and top box edges represent the first and third quartals, respectively, the dots above and below the boxes represent potential outliers and the last points at the bottom and at the top of each box represent the minimum and maximum values, respectively. The number of reactions per box plot is in Appendix Table S7. Source data are available online for this figure.

Moreover, similarly to the association with blood glucose levels, multiple nucleotides, amino acids, carbohydrates, and vitamins were found to have significantly lower diet-to-community fluxes in diabetic patients (Appendix Table S3). In contrast, nucleotides, amino acids, vitamins, and carbohydrates have increased within-community fluxes in diabetic patients (Fig. 2A; Appendix Fig. S2; Appendix Tables S2 and S3).

In summary, the metabolic fluxes among the microbiome are generally higher with increased blood glucose and in diabetic patients, and the metabolic fluxes from the diet to the microbiome are lower with increased blood glucose and in diabetic patients.

## Loss of microbial metabolites provided to the host in diabetes

Three community-to-host metabolic fluxes were found to be positively associated with blood glucose, and two were found to be negatively associated (Fig. 2B; Appendix Table S3). Notably, butyrate, a short-chain fatty acid, is produced at reduced levels by diabetic microbial communities ($P = 0.038$) (Fig. 2A). In addition, an increased production of H2S by the diabetic microbiome has been observed ($P = 0.001$) (Fig. 2C and Appendix Fig. S3). Moreover, the community-to-host fluxes of L-valine ($P < 0.0001$) and L-alanine ($P < 0.0001$) are significantly lower in diabetic individuals. The same applies to the Tn antigen ($P = 0.0051$), which is a monosaccharide linked to serine or threonine and plays a role in mucin glycosylation (Coletto et al, 2023). Furthermore, the production and consumption of metabolites associated with diabetes by individual genera was investigated. This was done for 18 genera, the five most abundant genera and genera showing a significant difference between healthy and diabetic individuals (Fig. 2C). Among the tested genera, there are nine butyrate producers, including *Clostridium*, *Lachnospiraceae* and *Eubacterium*. Those butyrate producers were found to have higher abundances in healthy individuals. Other genera include *Faecalibacterium* and *Campylobacter*, which are found to produce malate and valine, and three genera that are found to produce alanine, which are *Phascolarctobacterium*, *Micrococcus* and *Butyricicoccus*.

## Nucleotide, bile acid, B vitamins and amino acids exhibit altered fluxes in the diabetic microbiome

Six out of the eleven diet-to-community metabolites that are negatively associated with blood glucose are nucleotides, namely, deoxyadenosine,

adenosine, uridine, cytidine, deoxyguanosine, and guanosine. The flux for those metabolites from the host's diet to the community is also significantly decreased in diabetic patients (all P values < 0.0001). In contrast, the fluxes of guanosine monophosphate ($P = 0.0087$) and deoxyadenosine ($P = 0.0213$) within the community are positively associated with blood glucose. Coinciding with this, the fluxes of deoxyadenosine, cytidine, uridine, guanosine monophosphate, and uridine monophosphate are increased within the community of diabetic patients (P values, respectively: 0.001, 0.0201, 0.0171, <0.0001, <0.0001). In addition, in diabetic patients, there is an elevated flux of deoxyadenosine from the community to the host ($P = 0.0498$). Nucleotide metabolism plays a crucial role in energy metabolism, nucleic acid synthesis and repair, cell-to-cell communication and signal transduction in bacteria (Zakataeva, 2021). Nucleotides are also utilised in most metabolic pathways for energy exchange and regulation (Kilstrup et al, 2005). It is noteworthy that bile acids follow an opposite trend, where the diet-to-community fluxes of taurochenodeoxycholate and taurocholate increase in diabetes (P values: 0.0003, 0.0001). And those bile acids are deconjugated by the microbiome to cholate and taurine. The community-to-host fluxes of chenodeoxycholate are also increased in diabetes ($P < 0.0001$). But the within-community fluxes of cholate and 3-dehydrocholate are decreased in diabetes.

Furthermore, among diabetic individuals, B vitamins such as nicotinamide (B3), riboflavin (B2), thiamin (B1), and folate (B9) exhibit decreased diet-to-community fluxes (P values, respectively: <0.0001, 0.001, 0.0015, <0.0001), excluding niacin (another form of B3), which increases in diabetes ($P = 0.0345$). Meanwhile, vitamin K, in two forms, also increased within-community flux in diabetic microbiomes ($P = 0.0259$).

Similarly, the diet-to-community fluxes of amino acids, L-tryptophan and L-tyrosine are reduced in diabetic patients (P values: 0.0085, 0.0002), while the flux of L-glutamine is increased ($P = 0.0125$). The amino acids taurine (potentially derived from the conjugated bile acids), ornithine, and L-valine have increased within-community fluxes (P values: 0.0469, 0.0117, <0.0001), while L-valine and L-alanine have increased community-to-host fluxes in diabetic individuals (both P values < 0.0001).

## Energy production and core metabolic pathways are positively associated with elevated blood glucose

We summarised all the internal reactions from all bacteria within each microbial community and tested their association with the

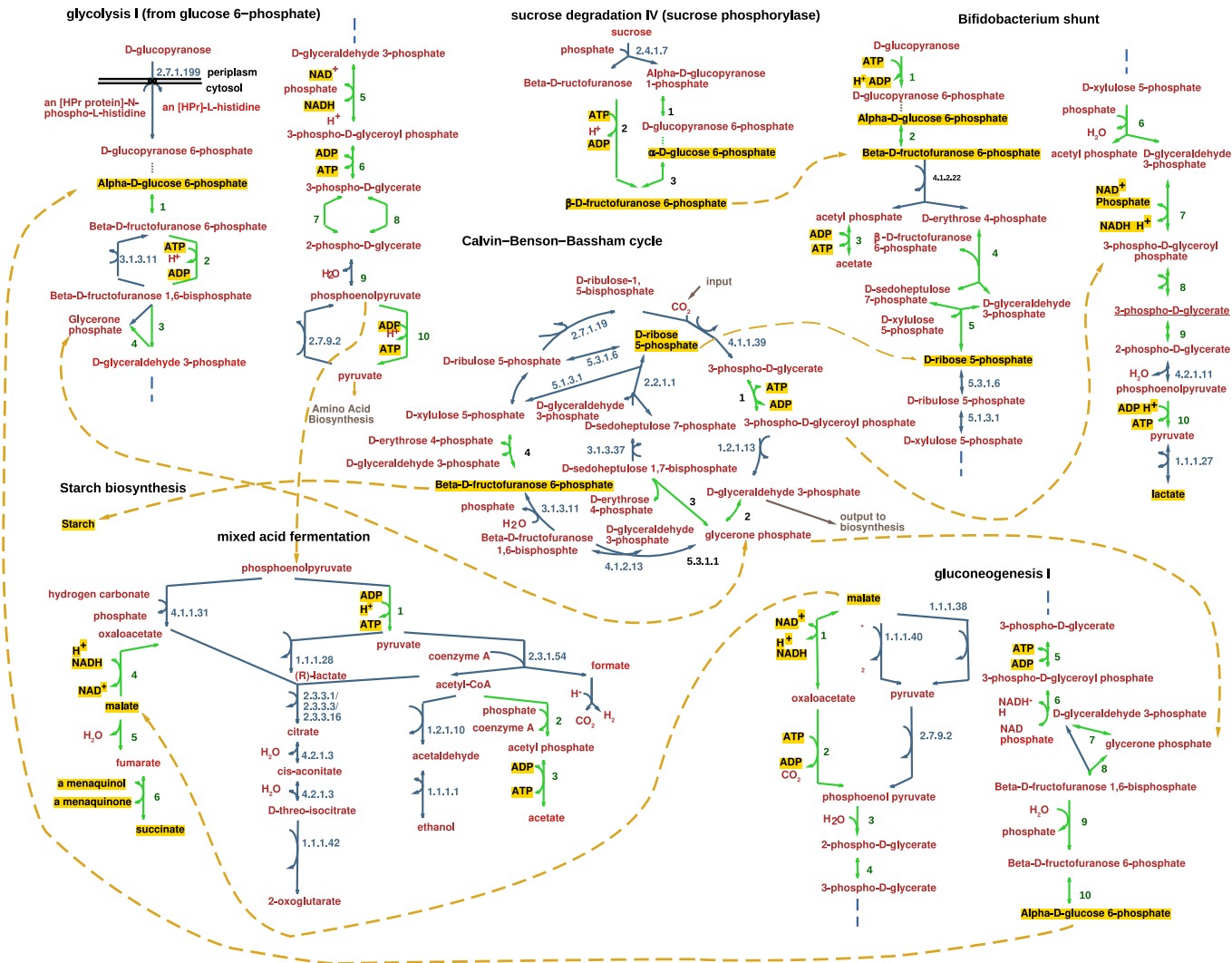

**Figure 3. Metabolic maps of subsystems enriched with reactions associated with blood glucose.**

An example of six major core metabolism pathways enriched with microbial reactions positively associated with the host's blood glucose. Arrows represent reactions, and two-sided arrows are used for reversible reactions. Green arrows represent reactions that are significantly associated with blood glucose, while blue arrows represent reactions that are not. Numbers next to green arrows are described in Appendix Table S7, while the ID numbers next to blue arrows are KEGG identifiers (Kanehisa and Goto, 2000). The metabolites are in red, and metabolites with fluxes that are associated with blood glucose, as well as those known to have connections to diabetes, are in black text with yellow highlights. Dashed orange arrows connect the pathways through shared metabolites.

blood glucose levels of the host. Starting with 4283 reactions, after filtering as described in the methods, we ended up with 855 reactions with a maximum of 840, a minimum of 517, and a median of 796 internal reactions per microbial community. By applying a linear model with confounders as described in the methods, we detected 507 reactions that are significantly associated with blood glucose (*P* values can be found in https://github.com/Sammerrr/Perfood). Considering the subsystem association of those reactions, we found 20 pathways that are significantly enriched with reactions that are associated with blood glucose using Fisher's exact test (Appendix Table S3). Most of those pathways belong to core metabolism, with the majority involved in energy metabolism.

We illustrated six major pathways in Fig. 3 based on EcoCyc metabolic maps (Keseler et al, 2021) to observe connections among

the enriched pathways and with the detected metabolites. Among the enriched pathways;

- Glycolysis I, which presents the first step in energy production from D-glucose by splitting it into pyruvates. This pathway results in the formation of ATP along with twelve intermediates that can be used as carbon skeletons for biosynthesis. Pyruvate in turn is used in amino acid I synthesis, gluconeogenesis as well in mixed acid fermentation.
- Gluconeogenesis I results in the biosynthesis of glucose from other non-carbohydrate sources, playing a role in energy production and regulation, and its balance with glycolysis maintains energy homoeostasis in bacteria.
- Mixed acid fermentation also has an important role in energy production. Glucose or other sugars are broken down

anaerobically to produce various acids while generating ATP and other compounds such as a-menaquinone, succinate, and malate. These can, in turn, be used by Gluconeogenesis I to produce D-glucose and maintain homoeostasis.

- Calvin−Benson−Bassham cycle is a central pathway that is also involved in energy production by using high-energy molecules to drive the production of glyceraldehyde-3-phosphate as well as β-D-fructofuranose-6-phosphate that enters the Starch biosynthesis pathway to produce starch.

- Sucrose degradation IV (sucrose phosphorylase), another pathway involved in energy metabolism, also produces β-D-fructofuranose 6-phosphate, which can enter the Bifidobacterium shunt.

- Bifidobacterium shunt in turn produces lactate, which can be used by butyrate-producing bacteria like some of the *Clostridium* genus to produce acetate and butyrate (Detman et al, 2019). Moreover, this pathway can produce D-ribose 5-phosphate which is essential for nucleotide biosynthesis.

# Discussion

In this study, we conducted a metabolic modelling-based analysis of changes in the gut microbiome associated with increased blood glucose levels and diabetes. To this end, we acquired 16S sequences along with metadata and dietary information from 1866 individuals, including 81 diabetic patients. It is important to highlight that participants in this study were not selected randomly; rather, they were individuals with a general interest in enhancing their health through personalised dietary recommendations (such as to lose weight or improve general fitness), which is a core aspect of Perfood's service. The imbalanced number of T2D patients in comparison to healthy individuals in this study presents a statistical challenge. To tackle this challenge, we attempted to isolate the statistical signal between T2D and the predicted metabolic fluxes, by correcting for confounders and adjusting for external variables that might bias the observed association. We chose confounders that are either strongly associated with T2D (e.g. carbohydrate intake) or, for some individuals, directly causal for T2D (e.g. measures of obesity (Finer, 2015)). Moreover, we also corrected for confounders that are known to influence the gut microbiome, such as lifestyle, gender and diabetes medication (Barlow et al, 2015).

The differential abundance analysis results are consistent with previous reports of an increase in *Streptococcus* and *Bacteroides* and a decrease in *Clostridium* among other butyrate-producing genera in diabetic children (de Goffau et al, 2014) and women (Karlsson et al, 2013). Multiple studies have reported that diabetes is usually accompanied by reduced gut microbial diversity (Hu et al, 2023; Chen et al, 2021). In this study, the observed increased richness in diabetic individuals with no difference in the other alpha diversity measures. This indicates that diabetic communities have a higher number of species but with low abundance, not resulting in an increased alpha diversity (Appendix Table S1). However, the simple measures of microbial diversity are limited in representing the actual changes in the community dynamics. Taxonomic diversity measures are unaware of the diversity on the genetic and metabolic levels; thus, they are susceptible to varying observations across cohorts (de Goffau et al, 2014; Karlsson et al, 2013) and are unable to detect the influence of important factors such as the difference

between antidiabetic medications (metformin or a combination of metformin and insulin) (Dzięgielewska-Gęsiak et al, 2022).

Moreover, we implemented a novel method to predict the different types of ecological interactions among the bacterial species in the gut microbiome and to estimate the relative frequencies of those interactions. The ecological interaction predictions in this study revealed a relative decrease in collaborative interactions (mutualism and commensalism) and an increase in exploitative interactions (antagonism and competition) in diabetic microbial communities. Such dysbiosis is potentially promoted by the high availability of simple energy sources such as glucose.

However, it is important to note that our analysis focused on changes in pairwise interactions determined from in silico predictions of isolated pairs of species. Thus, those interactions might differ in the context of the entire community.

Our investigation revealed extensive associations between blood glucose levels as well as clinically diagnosed diabetes and microbiome metabolic functions. Notably, all the metabolic fluxes that are positively associated with blood glucose were found to be significantly higher in diabetic patients, while negatively associated metabolites were significantly lower in diabetic patients. Elevated blood glucose was accompanied by increased fluxes of eleven metabolites among the microbial community members. Also, eleven metabolites showed a decrease in their flux from the host's diet to the microbial community (Fig. 2B). Among those metabolites, nucleotides, B vitamins, and amino acids had elevated fluxes within the community and decreased fluxes from the host's diet to the community accompanying higher blood glucose levels. This supports the notion of a diabetic microbiome that uses the vastly available glucose from the environment and becomes less controlled by the host and more dependent on the metabolic interchanges among its members.

With higher blood glucose, four metabolites, D-glucose, glutamine, lactate and fructose, showed higher diet-to-community flux. Those metabolites provide energy for the community and substantiate the connection between blood glucose levels and glucose availability to the gut microbiome. This agrees with the previously reported increase in the amounts of faecal carbohydrates, especially monosaccharides, in individuals displaying insulin resistance, where the elevated levels of monosaccharides were found to have a direct link to microbial carbohydrate metabolisms (Takeuchi et al, 2023).

Moreover, the observed reduction in the production of the short-chain fatty acid butyrate in the diabetic community is consistent with previous studies reporting a decreased abundance of butyrate-producing bacteria in diabetic patients (de Goffau et al, 2014; Hu et al, 2023; Chen et al, 2021; Qin et al, 2012). In turn, it coincides with reported decreased faecal butyrate concentrations in type 1 diabetic patients in comparison with healthy controls (Winther et al, 2021). Butyrate is also known to have an important role in moderating inflammatory responses (Chen and Vitetta, 2020; Amiri et al, 2022). In addition, it constitutes an essential source of energy for the colon and plays a signalling role by binding to the G-protein coupled receptors Gpr41 and Gpr43, which suppresses insulin signalling and prevent the accumulation of fat; therefore, butyrate helps to maintain the energetic balance (Tilg and Moschen, 2014). A reduction in the diet-to-community flux of L-tryptophan was also observed in this study. Tryptophan can be converted into indole by the bacteria that acts as a signalling

molecule and plays a role in regulating the integrity of the intestinal barrier, hence influencing liver health and systemic metabolic balance (Li et al, 2021), in addition to its anti-inflammatory role mediated by AhR signalling. Moreover, both tryptophan and indole have been reported to have an important impact on inflammation and insulin sensitivity (Ranhotra, 2023; Zhang et al, 2022). In addition, the increased production of H2S by the diabetic microbiome aligns with a report by Quin and others of an increase in sulfate-reducing bacteria in the diabetic microbiome (Qin et al, 2012). It is important to note that in normal amounts, H2S plays an important role in gut motility, sensing, secretion, and absorption, suggesting potential implications for gastrointestinal disease therapies. Another metabolite that was increased in diabetic microbial communities is sorbitol, which is produced by *Yokenella* and *Shigella* genera and was previously reported to have glucose intolerance-inducing effects (Li et al, 2022). These results highlight pathways in butyrate, tryptophan and H2S metabolism as potentially interesting targets for counteracting microbiome changes associated with T2D.

B vitamins that exhibit decreased diet-to-community fluxes in diabetic individuals act as coenzymes in various cellular reactions. They participate in all energy-producing processes within cells (Hossain et al, 2022; Putnam and Goodman, 2020). Despite the reduced uptake of these vitamins by the community, it is noteworthy that multiple bacteria can synthesise B vitamins. Coupled with the heightened energy metabolism in diabetic gut communities this supports the notion of a reduced influence of the host's diet on diabetic microbiomes.

Notably, all of the associations between the microbial internal reactions and blood glucose were positive. This is consistent with the observation that all the metabolic exchange fluxes among the community (within-community fluxes) are also positively associated with blood glucose levels (Fig. 2D). The observed enriched microbial pathways in diabetic patients are essential for energy metabolism, their increased activity in correspondence to high blood glucose levels further supports the notion that the microbiome becomes active in producing the energy it needs for its growth, starting with glycolysis that utilises glucose to the gluconeogenesis that maintains homoeostasis.

To conclude, our study employed a novel approach for predicting the ecological interactions between species pairs in microbial communities and further highlights the importance of metabolic modelling for the prediction of functional changes related to complex multifactorial diseases such as T2D. In light of our observations, we concluded that the availability of glucose as a simple source of energy, along with the disturbance of the community structure caused by diabetes, alters the metabolic relationship between the host and microbiome, causing a state of dysbiosis. The diabetic microbiome becomes less dependent on the host, less shaped by the host's diet, and more active in its energy production. Moreover, The ecological interactions among the community member species become more hostile with a shift towards exploitative interactions. Our study emphasises the necessity for comprehensively exploring the intricate interplay between diabetes and the gut microbial community. It provides an important basis for further experimental validations and dietary intervention analysis to explore how to potentially alter the dysbiotic effects of diabetes on the interactions between the microbiome and the host.

# Methods

## Reagents and tools table

| Reagent/ resource | Reference or source | Identifier or catalogue number |
|---|---|---|
| **Software** | | |
| USEARCH | Edgar, 2010 | |
| MicrobiomeGS2 | www.github.com/Waschina/ MicrobiomeGS2 | |
| EcoGS | https://github.com/maringos/ EcoGS | |

## Collection of metadata and 16S sequences

### Metadata

In this study, Perfood established a cohort dataset comprising 2117 individuals who purchased a Perfood product and provided consent for the anonymous use of their data for research purposes. Following the exclusion of cancer patients, individuals with IBD, those undergoing antibiotic treatment during sample collection, and those with repeated measures, the final dataset consisted of 1866 individuals, with each participant contributing a single sample of each type.

The cohort comprised 81 patients with type 2 diabetes and 1785 non-diabetic individuals. Clinical metadata, including gender, age, waist-to-hip ratio, physical activity, medication against diabetes and other diseases and clinical measures, were collected using questionnaires (Appendix Fig. S3). Physical activity was estimated by aggregating the daily duration across all types of physical activities the participants undertook. Baseline blood glucose levels (before meals) were estimated using a pipeline developed in-house by Perfood, patented and published (Twesten et al, 2021). This machine learning pipeline was trained using tens of thousands of baseline glucose values set by medical diabetologists, as blood glucose values before a meal vary and heavily rely on the correctness of the start time of the meal. In addition, HbA1c levels were collected using continuous glucose monitoring (CGM) sensors and averaged for the aim of this study. Microbiome and metadata for the human cohort can be obtained upon request to Perfood GmbH.

### Sequence data

Faecal samples were obtained from each participant and processed as described by Kordowski and others (Kordowski et al, 2022). The 16S rRNA gene sequences were acquired using Illumina paired-end short-read sequencing. Prior to downstream analysis, adaptor sequences were trimmed, and the 16S sequences were filtered for host sequence contamination and screened for errors, where regions with high variability in k-mer frequencies, indicative of potential sequencing errors, were identified. The majority of the 16S sequences exhibited low error rates, less than 0.05, and sequences with higher error rates were excluded. Subsequently, the forward and reverse sequences were merged. These steps were performed using the R package dada2 (Callahan et al, 2016). A list of all functions utilised is available in Appendix Table S4.

### Consent

The 16S rRNA sequencing data was generated in the context of a prior project (University of Lübeck, ethics notification ID 20-415) (Kordowski et al, 2022). All participants gave their consent to the general use of their anonymised data for scientific purposes. Sequencing data and metadata were provided completely anonymised and thus did not require separate ethical approval according to national regulations for the analysis presented in this manuscript.

### Diet estimation

The total amounts of proteins, carbohydrates and fats consumed by the participants (in grams) as well as the caloric intake at the time preceding the sample collection were estimated from surveyed dietary information per participant. In order to estimate the expected frequencies of macro- and micro-nutrients (sugars, amino acids, fatty acids, vitamins and ions) present in the diet, we used a reference diet that is already curated and adjusted to ModelSEED namespace (Pryor et al, 2019). The underlying ingredient database can be obtained through the FoCus cohort (PopGen Biobank, Schleswig-Holstein, Germany) and can be accessed via a structured application procedure (https://www.uksh.de/p2n/Information+for +Researchers.html).

The reference diet contained detailed dietary element information for 2396 individuals.

For each individual from the reference diet, we determined the proportion of each amino acid to total proteins, the proportion of fatty acid to total fat, sugar to total carbohydrates, and micronutrients (vitamins or minerals) to the caloric content. Thereafter, these proportions were then averaged across all participants in the reference diet in order to establish reference ratios and eliminate individual biases (Appendix Table S5). Subsequently, this reference nutritional element ratios were multiplied by the total amounts of proteins, carbohydrates, fats and caloric intake estimated from our cohort. Thus, we were able to estimate the amounts of micro- and macronutrients for each participant in our cohort (in grams per day). The estimated dietary element amounts were converted into fluxes per day, mMol per bacterial weight in the human gut.

$$Flux = \frac{\frac{EW}{M} \times 1000}{gmW}$$

Where:

$eW$: The estimated intake for a dietary element in grams.

$MW$: Molecular weight in mMol/g.

$gmW$: Estimated gut microbiome weight = 200 g (Sender et al, 2016).

The resulting fluxes were normalised by caloric intake and integrated into the community models by setting the lower bounds of the community models of the corresponding dietary element to its predicted flux per participant.

### Microbial diversity and ecological interactions

The 16S sequences were mapped against a reference collection of the human gut microbiome (AGORA) (Magnúsdóttir et al, 2017) using the USEARCH software (Edgar, 2010). We used the whole genomes of bacteria from the AGORA collection identified by the mapping procedure to construct genome-scale metabolic models using gapseq

(Zimmermann et al, 2021). We did not use the original AGORA metabolic reconstructions since they only poorly grow with the molecular composition of a typical northern German diet (Pryor et al, 2019). Upon mapping the 16S sequences against the AGORA reference collection, we calculated the abundances of individual bacterial species within each microbial community. This was achieved by counting the 16S sequences that mapped to a certain genome for each participant and dividing this number by the sum of all 16S sequence counts within the community. We thereafter removed bacterial species with low abundances (below 0.1%) from the communities. This resulted in 499 bacterial species belonging to 148 genera that are present in different abundances within the gut microbial communities of the participants. Three alpha diversity measures, namely, the Shannon index, Chao1 index and species richness, as well as beta diversity, were estimated for the microbial communities using the vegan R package (Oksanen J et al, 2022).

To predict the ecological interactions between each possible pair of the identified microbial species ($n = 499$ species) among the communities, we started by creating a half matrix of 124,251 species pair combinations ($\frac{n \times (n-1)}{2} = 124,251$). For each of those species pairs, we applied our self-developed R package EcoGS. This package employs metabolic modelling (FBA) to compare the predicted growth of bacterial models with their co-growth after merging them into a community model using Sybil (Gelius-Dietrich et al, 2013) and MicrobiomGS2 (Waschina, 2019) R packages, respectively. The EcoGS package is available and described in more detail on GitHub (https://github.com/maringos/EcoGS). To that end, we categorised each bacterial species pair into one of the possible six ecological interactions (Appendix Table S6). Thereafter, we counted the number of pairs belonging to each ecological interaction in each microbial community and weighed those counts by the abundance of the species in each pair as follows:

Given that the abundances of the species s1, s2, s3, s4, s5, etc. are: a1, a2, a3, a4, a5, etc.

For a certain type of ecological interaction $E$, we have $NE$ number of pairs of species that belong to that type of interaction:

(s1, s2), (s1, s3), (s3, s5), etc.

And the weighted estimate of that interaction ecological interaction: $WNE = (a1 \times a2) + (a1 \times a3) + (a3 \times a5) + \dots$

One effective way to account for the codependency within compositional data is to use a logarithmic transformation of the ratio between every two variables (each two ecological interaction percentages in our case) (Greenacre, 2021). The six possible ecological interactions between bacteria resulted in 15 combinations of log ratios (log (Antagonism/Competition), log (Commensalism/Mutualism), … etc.).

## Microbial community modelling

The estimated growth of all individual genome-scale species models exceeded the threshold of flux $<10^{-3}$ on the modelled diet (flux units are arbitrary). Hence, all the species models were included in the community modelling. Constraint-based modelling was implemented as flux balance analysis (FBA) using the coupling constraints described in (Heinken et al, 2013) to simulate the community metabolic models using MicrobiomeGS (Aden et al, 2019) and in-house scripts. The bacterial models that belong to each single community were merged into a community model, where each bacterium has its own compartment, and the community members exchange metabolites

freely among each other and with the host in an environmental compartment. Thereafter, a community-level biomass reaction was introduced that accounts for the individual bacterial biomasses according to their relative abundance in the respective sample. The lower bounds of the exchange reactions were adjusted individually according to the estimated diet for each participant. An objective function was set to optimise the community's growth. The cplexAPI R package (Gelius-Dietrich, 2020) was used to optimise growth using the linear solver IBM ILOG CPLEX through its R interface with an academic license (Cplex, I.I. 2009. V12.1).

Thereafter, metabolites and reactions were removed if their fluxes had a low standard deviation ($< 0.01$) among the different communities or if they were only present in less than 70% of the diabetic microbial communities.

## Statistical analysis

The strong correlation between baseline glucose and HbA1c ($P$ value $< 2.2e\text{-}16$, rho $= 0.96$) led us to consider a particular measure in this study associated with blood glucose levels if it is correlated with either fasting glucose, HbA1c, or both.

All statistical analysis was conducted with R, version 4.2.3 (R Core Team, 2023) within the RStudio environment (Posit team, 2023). The plots were created with the ggplot2 R package (Wickham, 2016). and R base plotting functions. The results of all statistical tests were determined to be significant below a threshold of $\alpha = 0.05$. $P$ values were adjusted for multiple testing using false discovery rate correction (FDR) (Benjamini and Hochberg, 1995). We opted to correct for confounders strongly associated with T2D (carbohydrate dietary intake and waist-to-hip ratio measure of obesity (Finer, 2015)), as well as gender, age, physical activity and, if applicable, the antidiabetic medication treatment.

The linear model formulas used in this study are as follows,
For baseline (fasting) glucose levels and HbA1c:

$$lm\,(blood\ glucose \sim metabolic\ flux + Gender + age \\ + waist\ to\ hip + activity + carbohydrate)$$

For diversity measures:

$$lm\,(alpha\ diversity \sim diabetes + Gender + age + waist\ to\ hip \\ + activity + anti - diabetics\ medication)$$

$$lm\,(alpha\ diversity \sim diabetes + Gender + age + waist\ to\ hip + activity)$$

For ecological interactions:

$$lm\,(blood\ glucose \sim ecological\ interaction\ ratio + Gender \\ + age + waist\ to\ hip + activity + carbohydrate)$$

No blinding was necessary in this study, as the data were initially collected from the participants in the context of a service product and were pseudonymised before the analysis.

## Data availability

The datasets and computer code produced in this study are available in the following databases: 16S sequences and clinical data, upon request from perfood GmbH (correspondence to Oliver Witt:

oliver.witt@perfood.de). Upon request, the data can be obtained within 60 days. Due to German national regulations, a deposition of data in a data access-controlled publicly available repository was not possible; Samples and data for the FoCus cohort were provided by the PopGen Biobank (Schleswig-Holstein, Germany) and can be accessed via a structured application procedure (https://www.uksh.de/p2n/Information+for+Researchers.html. Upon application, the FoCus data can be obtained within 7–30 days. Computer scripts, flux exchange prediction data and statistical results: GitHub (https://github.com/Sammerrr/Perfood).

The source data of this paper are collected in the following database record: biostudies:S-SCDT-10_1038-S44320-025-00100-w.

## Peer review information

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

## Acknowledgements

The authors acknowledge support by the German Research Foundation within the framework of the Excellence Cluster "Precision Medicine in Chronic Inflammation" (project code EXC2167) and the project ExoMod (DFG support code KA 3541/20-1). The authors thank Jan Taubenheim, Karlis Arturs Moors, Silvio Washina, Johannes Zimmermann, and Robin Koch for fruitful discussion and critical comments.

## Author contributions

**A Samer Kadibalban**: Conceptualisation; Software; Formal analysis; Visualisation; Methodology; Writing—original draft; Writing—review and editing. **Axel Künstner**: Conceptualisation; Data curation; Investigation. **Torsten Schröder**: Conceptualisation; Data curation; Funding acquisition; Investigation. **Julius Zauleck**: Data curation; Investigation. **Oliver Witt**: Data curation; Investigation. **Georgios Marinos**: Software. **Christoph Kaleta**: Conceptualisation; Funding acquisition; Methodology; Project administration.

Source data underlying figure panels in this paper may have individual authorship assigned. Where available, figure panel/source data authorship is listed in the following database record: biostudies:S-SCDT-10_1038-S44320-025-00100-w.

## Funding

## Disclosure and competing interests statement

Perfood, founded in 2017, is a MedTech startup that develops medical and lifestyle products aimed at preventing or treating diseases, as well as facilitating healthy lifestyle changes. The core approach is based on precise monitoring of lifestyle factors through the use of sensors, microbiome data and user input, enabling the delivery of personalised recommendations. The data used in this publication comes exclusively from Perfood's lifestyle product, MillionFriends. The co-authors, Axel Künstner, Torsten Schröder, Julius Zauleck, and Oliver Witt, are affiliated with Perfood GmbH. The authors declare no competing or commercial interests. The data for this study were collected up to 2021, and the microbiome data is not part of any product by Perfood GmbH as of today.

