## [Peer Review File · Molecular Systems Biology]

Metabolic modelling reveals increased autonomy and antagonism in type 2 diabetic gut microbiota

A. Samer Kadibalban, Axel Künstner, Torsten Schröder, Julius Zauleck, Oliver Witt, Georgios Marinos, and Christoph Kaleta

Corresponding author(s): Christoph Kaleta (c.kaleta@iem.uni-kiel.de)

Review Timeline:

Transfer from Review Commons:	10th Jan 25
Editorial Decision:	20th Jan 25
Revision Received:	11th Mar 25
Accepted:	31st Mar 25

Review
COMMONS

Editor: Poonam Bheda

Transaction Report:

Review #1

1. Evidence, reproducibility and clarity:

Evidence, reproducibility and clarity (Required)

Summary:

In this manuscript, Kadibalban et al. explore the metabolic ecological interactions in type 2 diabetes (T2D), including diet-to-microbiome, intra-microbiome, and microbiome-to-host interactions. They characterize the microbiome of a cohort of over 1800 individuals, generate metabolic network models based on the microbiome characterizations, and integrate these models into various community modeling simulations to identify types of ecological interactions, metabolites, and pathways that differ between healthy and sick individuals. The authors identify that there are fewer diet-to-microbiome interactions and more negative intra-microbiome ecological interactions in T2D, along with identifying several metabolites and pathways of interest, including lowered butyrate contribution to host and dietary tryptophan to the microbiome. The described impact of this work is metabolites suggested as treatment for T2D. Overall, this study demonstrates a novel framework through which to assess diet-host-microbe interactions using metabolic network modeling and patient data. Specific comments and feedback follow.

Major Comments:

1. The authors suggest that the findings reported here could be used as treatments for T2D (Lines 475-476), however there is limited support for this claim within the experiments described in the manuscript. The authors should consider that while the microbiome is implicated in T2D, it is not known to be causal. As such, restoring some functionality of the microbiome through intervention of metabolites may not reduce insulin resistance or promote the secretion of insulin but rather may be more likely to reduce GI sequalae associated with T2D.

2. The framing of the paper does not highlight its greatest impact. The authors chose to frame the paper as revealing novel biology in T2D, when there was very little validation, the T2D cohort was less than 10% of the study group, and the impact on future research directions is unclear (See Major Comment 1). A more effective framing for this study would be on the modeling approach used to assess the host-microbe interactions and intra-microbiome changes occurring in microbiomes, along with functionally exploring a study group that has dynamics that are not always fully captured by diversity measurements

(PMID: 36426212, PMID: 36426212). The connection to T2D is weak, but the approach and framework used to do the simulations is highly novel and could lead to major advances in understanding microbiome interactions.

3. The authors mention that they used EcoGS to perform their pairwise simulations, but provide very limited details on this package, instead instructing readers to visit the GitHub repository (Lines 157-172). This GitHub repository does not contain detailed information regarding the assumptions and methods of this simulation, and only notes that a manuscript describing the package is currently in preparation. The authors should describe this novel method more clearly and completely; currently, neither the GitHub nor manuscript describe the method sufficiently.

4. The authors might consider that bacterial pairs in a microbiome may have different interactions with each other within the context of a larger ecosystem. EcoGS might predict pairwise interactions, but that same pair might interact differently in the community setting.

5. The authors should include all raw and analyzed 16S data, metadata files, and code as supplement or deposited in a publicly available repository. Without these components, reproducibility of the study is limited.

6. The dietary components of the cohort were determined using a proprietary nutrient database (Line 131). The disclosure of this portion of the database would be critical for reproducing the exchange bounds on the models.

****Minor Comments:****

1. In Section 3.2 (Lines 144-154) The authors do not explain why 16S needed to be mapped to AGORA to identify species and genera, but they opted to use GapSeq rather than the already existing AGORA models. Upon consulting the EcoGS GitHub, it became clear that this pipeline only accepts GapSeq generated models. Being clear about this would eliminate confusion regarding the model generation method.

2. Since the cohort of T2D patients is less than 10% of the total study population, the choice of confounders may be correcting for the effect of T2D on these microbiomes (Lines 198-199). The authors corrected for variables either strongly associated with T2D (e.g. carbohydrate intake) or, for some individuals, directly causal for T2D (e.g. measures of obesity). Citing other studies from the T2D literature supporting the choice of T2D-

associated "confounders" would be helpful.

3. The difference between the number of T2D subjects vs healthy controls is concerning, as T2D represents less than 10% of the cohort. This could impact statistical analyses describing differences between healthy and sick, and the authors should at least acknowledge this limitation.

4. The authors report that they removed species with low growth rates from the community metabolic simulations. However, these kinds of slow-growing species have critical metabolic contributions to the overall community (e.g. acetogens). It would be helpful to provide information on what species were removed from the simulations and how the simulations changed before and after removing the slow-growing community members.

5. The presentation of taxa-based results is not informative. Based on the information provided in the Results section, if the raw 16S data were available, a reader would still have to perform their own analysis to get useful information, like taxa bar charts and lists of identified taxa. It would be helpful if Figure 1 contained a new panel summarizing the 16S data in a taxa bar chart and if the authors provided a more extensive summary of the findings in the Results section beyond listing numbers of identified taxa.

6. When describing the simulation results, the authors just provide numbers of identified fluxes. Since the models are not included in the supplement, it would be helpful to add a supplemental file with a list of all potential metabolite fluxes, which ones were identified in which simulations, and which were trimmed in the analysis pipeline.

7. Grammatical Issues

a. Lines 59-62: Run-on sentence that requires rewording for clarity

b. Lines 78-79: Consistency in plurals

Example correction: "Genome-scale metabolite models are powerful tools that provide comprehensive representations of entire metabolic networks of organisms. These models..."

c. Line 198: "gender" instead of "Gender"

d. Line 211: "Although," instead of "although"

e. Line 296: "butyrate" instead of "Butyrate"

f. Line 298: "alanine" instead of "Alanine"

g. Lines 322-326: Font and text size inconsistent with rest of manuscript

h. Lines 342-366: Font and text size inconsistent with rest of manuscript

2. Significance:

Significance (Required)

This paper serves as an intriguing and novel exploration of ecological interactions between host, diet, and microbiomes. Most notably, the integration of human microbiome 16S data to inform community models adds strength to this study. In addition, the literature has previously reported lack of significant difference in diversity measurements between T2D and healthy microbiomes, and this study begins to look at functional differences in this disease that go beyond diversity measures associated with more clinically-focused microbiome studies. The *in silico* approach to modeling diet-to-microbiome and microbiome-to-host interactions is a nuanced and novel approach. Although there are some concerns with reproducibility of the study given lack of details on EcoGS, lack of availability of raw data, and certain metrics that were calculated using proprietary approaches, this paper adds novel contributions to modeling host-microbiome interactions and elucidating biological mechanisms that drive functional changes in microbiomes. The audience for this manuscript would be somewhat specialized for those who have experience with *in silico* metabolic modeling and studying microbial communities. Those involved in translational research might be particularly interested in the work, since a large human cohort is utilized, and potential biomarkers of microbiome dysfunction are identified in the context of a human metabolic disorder. Translational investigators might also use some of the identified pathways and metabolites to alleviate GI sequelae associated with T2D. Those with specific interest in the modeling of microbial communities would also find use in the methods presented as an approach to model host-microbiome interactions. I am a systems biologist, and my field of expertise involves metabolic networking modeling of microbial communities, with an emphasis in gut microbial communities. Although I have limited expertise in the field of T2D specifically, I am well versed in the study of the gut microbiome and in the ways that these communities have been modeled *in silico*.

3. How much time do you estimate the authors will need to complete the suggested revisions:

Estimated time to Complete Revisions (Required)

(Decision Recommendation)

Between 1 and 3 months

Yes

Review #2

1. Evidence, reproducibility and clarity:

Evidence, reproducibility and clarity (Required)

In this study, the authors aim to describe the difference of healthy and diabetic gut microbiota in terms of metabolic modeling. They quantify and compare ecological interactions of microbes in different hosts. Moreover, they use estimated metabolic fluxes to predict the blood glucose level of patients.

Although the authors compiled an impressive amount of raw data, their methodology is poorly explained and thus, in my opinion, their conclusions are not supported well enough. I am especially skeptical about the "precision diet" estimated by the authors. Not only is the method section way too under-detailed to be ever replicated, but also the results of the diet fluxes are not reported. As community models depend on the values of these dietary fluxes, not reporting them is a critical oversight.

Moreover, I am not confident on the baseline blood glucose levels estimation and manual curation, which is not explained and only a non-peer reviewed patent is cited. Here also a more detailed methodology section would increase the quality of the study significantly. Finally, no code of the analysis (GitHub, ...) and no raw data has been published alongside this paper, which is a must for theoretical analyses.

****Major Points:****

Line 114f: As this glucose estimation pipeline is not peer-reviewed, in my opinion, it is fair to include a detailed explanation in here.

* To what extent was the manual curation performed?

Line 128f, Precision Diet: In my opinion this paragraph is way too under-detailed. Methods should be repeatable. This is not the case here.

* What is actually measured in the Perfood digital diary platform?

* What does this in-house dietary ingredient database comprise? Please publish the database in the supplementary material.

* What does the "reference precision diet" comprise out of? Why is it "precise"?

* What does frequency mean in this context? If it is not per time unit, maybe another word is more suitable.

* What is and how big are FOCUS and Perfood cohorts?

* Why is averaging performed?

Considering all the steps necessary to get the estimate, I suggest completely removing the "precision" claim, as I cannot see how this is true.

I find the term "precision" especially hard to believe as you cannot even report units to your fluxes.

Line 161f: Here I am missing many points that are explained in Section 3.3. Are they the same? If yes, it may be helpful to refer to them.

* What is the concrete objective function?

* How are different microbial abundances handled?

* Are the microbes in two different compartments?

* How do you deal with non-unique solutions?

Line 233:

* Is this an increase due to an increase in blood glucose?

* Why should there be different levels of blood glucose in healthy subjects?

* If "healthy" subject have too high glucose baselines are they really healthy or just not diagnosed?

Section 4.3:

* For me it is not clear if you talk about all models or just a single one.

* Did it happen that for certain samples not all exchange reactions you were interested where present? How did you deal with that?

Section 4.5:

* Line 287: Please add where I can see the H₂S production.

* Please at p-values when you report significance

Line 469:

* Are there sulfate-reducing bacteria in your dataset?

General:

* Please report the significance/ p-value of your regression parameters

* For me it is not clear, how the linear models are constructed. What is your independent/ what is your dependent variable? Why is this not part of the methods section? Please explain in more detail.

Line 437:

* As high glucose is the main symptom of diabetes, I do not understand how a correlation between fluxes predicting high glucose and being high in diabetics proves any concept.

****Minor Points:****

- Line 84: IBD abbreviation not defined

- Line 103: As the Perfood company is so central to this study, please give a small overview of Perfood what they do and what "Perfood product" and "Perfood digital diary platform" is supposed to mean.

- Line 167: I do not understand the mathematical operations performed, maybe an Equation is warranted.

- Line 158: Please cite the "logarithmic transformation" study.

- Line 176: what is meant by coupling constraints?

- Line 187: Filtered out of what?

- Line 197: Please cite the FDR correction used.

- Line 198 (and others): Is the confounder gender or sex?

1) Fig 1, Tab S3:

* Indicate P-value levels (*, **, ***) as common in scientific publications.

* It would be nice that the significant p-values are part of the text (Line 213) and fig 1 ACD.

* In Fig 1A, D you write fold difference, but in the caption you write $\log_2(\text{fold difference})$ please clarify.

* The categorical coloring of Fig 1A does not work well with the continuous data presented. Please reconsider.

* Fig 1B, as the frequencies are so different in size, please use a log scale on your Y-axis.

* Are the interaction ratios of CD also log-scale as reported in the methods? If yes, please indicate!

2) Fig 2:

* Please explain what AC, CtH, HtC is. In panel A red is community-to-host, but in Panel B it is "HtC" which may mean host-to-community. What is correct?

* Panel B: please write $\log(\text{LM estimate})$ & report either p-values or at least the significance levels. Also the equation is not displayed correctly.

* Panel A: Number of *metabolic fluxes* not metabolites?

3) Spelling: CO₂, H₂S, H⁺ all should have the correct subscript.

4) Section 4.4 is missing

2. Significance:

Significance (Required)

I think that the authors have a great dataset at hand. However, the methodology section does not suffice for repeatability of the results.

The authors show that a metabolic disease of the host also affects their microbiome and highlights the importance of microbiome-host interactions during illness.

3. How much time do you estimate the authors will need to complete the suggested revisions:

Estimated time to Complete Revisions (Required)

(Decision Recommendation)

Between 1 and 3 months

Yes

Full Revision

Manuscript number: RC-2024-02566R

Corresponding author: Christoph Kaleta

1. General Statements

Dear Dr. del Álamo, Dear Editors:

We thank you and the reviewers for their valuable suggestions and helpful comments on our manuscript. We appreciate the suggestions of the reviewers and believe that the quality of our manuscript increased clearly with the corresponding revision.

The main points that we have addressed are:

We have expanded explanations for the EcoGS software and dietary modeling pipeline, addressing assumptions, methods, and limitations (e.g., handling microbial abundances, dietary flux estimations, and glucose pipeline manual curation)

We have improved reproducibility by publishing metabolic flux tables, and analysis code on GitHub, have added access details for proprietary databases and clarified limitations in sharing metadata due to privacy concerns.

We now acknowledge limitations such as the small T2D cohort size and confounder effects, providing citations to support the choice of confounders.

We have revised analyses to ensure robustness, such as confirming that no slow-growing species were removed and clarifying the construction of linear models and p-value reporting

Please find enclosed our revised manuscript and a point-by-point response to the reviewers' comments. Please do not hesitate to contact us in case of further questions.

Yours faithfully,

Samer Kadibalban and Christoph Kaleta

Reviewer #1 (*our responses in italics*)

Evidence, reproducibility and clarity (Required)

Summary:

In this manuscript, Kadibalban et al. explore the metabolic ecological interactions in type 2 diabetes (T2D), including diet-to-microbiome, intra-microbiome, and microbiome-to-host interactions. They characterize the microbiome of a cohort of over 1800 individuals, generate

metabolic network models based on the microbiome characterizations, and integrate these models into various community modeling simulations to identify types of ecological interactions, metabolites, and pathways that differ between healthy and sick individuals. The authors identify that there are fewer diet-to-microbiome interactions and more negative intra-microbiome ecological interactions in T2D, along with identifying several metabolites and pathways of interest, including lowered butyrate contribution to host and dietary tryptophan to the microbiome. The described impact of this work is metabolites suggested as treatment for T2D. Overall, this study demonstrates a novel framework through which to assess diet-host-microbe interactions using metabolic network modeling and patient data. Specific comments and feedback follow.

Major Comments:

1.1. The authors suggest that the findings reported here could be used as treatments for T2D (Lines 475-476), however there is limited support for this claim within the experiments described in the manuscript. The authors should consider that while the microbiome is implicated in T2D, it is not known to be causal. As such, restoring some functionality of the microbiome through intervention of metabolites may not reduce insulin resistance or promote the secretion of insulin but rather may be more likely to reduce GI sequelae associated with T2D.

We thank the reviewer and agree with the comment, we changed the statement in the manuscript to: "These results highlight pathways in butyrate, tryptophan and H2S metabolism as potentially interesting targets for counteracting microbiome changes associated with T2D". Lines 541 and 543.

1.2. The framing of the paper does not highlight its greatest impact. The authors chose to frame the paper as revealing novel biology in T2D, when there was very little validation, the T2D cohort was less than 10% of the study group, and the impact on future research directions is unclear (See Major Comment 1). A more effective framing for this study would be on the modeling approach used to assess the host-microbe interactions and intra-microbiome changes occurring in microbiomes, along with functionally exploring a study group that has dynamics that are not always fully captured by diversity measurements (PMID: 36426212, PMID: 36426212). The connection to T2D is weak, but the approach and framework used to do the simulations is highly novel and could lead to major advances in understanding microbiome interactions.

We also thank the reviewer for the point of view and the suggested citation, indeed, by the time of submitting our manuscript, there have been limited attempts to use metabolic modelling to study the differences between diabetic and healthy individuals' gut microbiomes on the predicted functional level and the ecological interaction levels.

We highlighted this more in the discussion and added the suggested citation (lines 483 to 491): "However, the simple measures ... the relative frequencies of those interactions. "

And in section 6, concluding remarks (lines 559 to 561):

"Our study employs a novel approach for predicting the ecological interactions between pairs in microbial communities and further highlights the importance of metabolic modelling for the prediction of functional changes related to complex multifactorial metabolic diseases such as T2D"

Furthermore, we would like to highlight that in this study, not only do we find associations of the predicted metabolic fluxes with diabetes but also with blood glucose levels, which adds an extra step towards the understanding of the complex relation between diabetes and the gut microbiome.

1.3. The authors mention that they used EcoGS to perform their pairwise simulations, but provide very limited details on this package, instead instructing readers to visit the GitHub repository (Lines 157-172). This GitHub repository does not contain detailed information regarding the assumptions and methods of this simulation, and only notes that a manuscript describing the package is currently in preparation. The authors should describe this novel method more clearly and completely; currently, neither the GitHub nor manuscript describe the method sufficiently.

We thank the reviewer for this important comment.

We added the following technical details to the software description in the GitHub repository (<https://github.com/maringos/EcoGS>):

"We used MicrobiomeGS2 as the basis for EcoGS, in each case, two bacterial models were merged into one and regarded as interconnected distinct compartments of the merged model. Importantly, we kept the objective function of each model independent from the other and optimised each objective simultaneously. The objective function was the biomass reaction of each model. Therefore, we did not assume any abundance but we inferred it from the

optimization of each biomass reaction. Depending on both resulting growth rates, we estimated the ecological interactions.

Although there are non-unique solutions to such optimization problems, the sybil software in R and the CPLEX solver return stable results. Therefore, in practice, we did not face this issue”.

1.4. The authors might consider that bacterial pairs in a microbiome may have different interactions with each other within the context of a larger ecosystem. EcoGS might predict pairwise interactions, but that same pair might interact differently in the community setting.

We thank the reviewer for this suggestion. Indeed, interactions might change in the context of a larger community. Yet, while it would in principle be possible to look at higher-order interactions, due to combinatorics this is computationally not feasible even for tripartite interactions (which would be roughly 89 million for all strains from the AGORA gut bacteria collection). We now mention this as a limitation in the discussion section of our manuscript (l. 498 - 500):

“However, it is important to note that our analysis focused on changes in pairwise interactions determined from in silico predictions of isolated pairs of species. Thus, those interactions might differ in the context of the entire community.”

1.5. The authors should include all raw and analyzed 16S data, metadata files, and code as supplement or deposited in a publicly available repository. Without these components, reproducibility of the study is limited.

We thank the reviewer for this suggestion. While we in principle agree with the notion that all data should be publicly available, there are some limitations concerning human data, as is normal for most human cohorts. Sharing the 16S data and metadata on the human cohort can not be shared publicly due to data privacy restrictions. However, data can be obtained through contact with Perfood GmbH. We have added the following statement to the data availability section of our manuscript (3.1 line 120)

“Microbiome and metadata for the human cohort can be obtained upon request to Perfood GmbH.

We have made all code, species abundance per participant, as well as all the metabolic simulation result tables available under the following link:

<https://github.com/Sammerrr/Perfood>

1.6. The dietary components of the cohort were determined using a proprietary nutrient database (Line 131). The disclosure of this portion of the database would be critical for reproducing the exchange bounds on the models.

We added the following statement in lines 138 to 140:

“The underlying ingredient database can be obtained through the FoCUS cohort (PopGen Biobank, Schleswig-Holstein, Germany) and can be accessed via a structured application procedure (<https://www.uksh.de/p2n/Information+for+Researchers.html>)”

Minor Comments:

1.7. In Section 3.2 (Lines 144-154) The authors do not explain why 16S needed to be mapped to AGORA to identify species and genera, but they opted to use GapSeq rather than the already existing AGORA models. Upon consulting the EcoGS GitHub, it became clear that this pipeline only accepts GapSeq generated models. Being clear about this would eliminate confusion regarding the model generation method.

Indeed, we used our own metabolic reconstructions of strains also included in the AGORA database. While it is true that EcoGS is somewhat specific for gapseq reconstructions, we chose to use our own reconstructions with gapseq since we found that bacteria from the AGORA collection were quite adapted to the predefined Western diet defined by the Thiele lab but did grow only poorly with a typical Northern German diet that we defined based on dietary questionnaires. For instance, in a previous work in which we used a Northern German diet in combination with AGORA reconstructions, we still had to provide 10% of the predefined Western diet to make them grow properly (Pryor et al., 2019). We added the following statement to the methods sections (164 - 166):

“We did not use the original AGORA metabolic reconstructions since they only poorly grow with the molecular composition of a typical Northern German diet (Pryor et al., 2019).”

1.8. Since the cohort of T2D patients is less than 10% of the total study population, the choice of confounders may be correcting for the effect of T2D on these microbiomes (Lines 198-199). The authors corrected for variables either strongly associated with T2D (e.g. carbohydrate intake) or, for some individuals, directly causal for T2D (e.g. measures of obesity). Citing other studies from the T2D literature supporting the choice of T2D-associated "confounders" would be helpful.

Full Revision

We thank the reviewer for this point.

We added the following text to the discussion lines 464 to 472:

“The imbalanced number of T2D patients in comparison to healthy individuals in this study presents a statistical challenge. We attempted to tackle this challenge by correcting for confounders in the statistical tests to adjust for external variables that might bias the observed association. Hence, to isolate the statistical signal between T2D and the predicted metabolic fluxes. We chose confounders that are either strongly associated with T2D (e.g. carbohydrate intake) or, for some individuals, directly causal for T2D (e.g. measures of obesity (38)). Moreover, we also corrected for confounders that are known to influence the gut microbiome, such as lifestyle and medication, including diabetes medication (49)”

We cite a study from T2D literature supporting the choice of confounders in discussion line 468. We also added a new citation providing evidence of the association between obesity and T2D in line 227 and line 466.

1.9. The difference between the number of T2D subjects vs healthy controls is concerning, as T2D represents less than 10% of the cohort. This could impact statistical analyses describing differences between healthy and sick, and the authors should at least acknowledge this limitation.

We added a statement in the discussion acknowledging this limitation and how we tried to counteract its effects by correcting for confounders (please see our response to comment 1.8).

1.10. The authors report that they removed species with low growth rates from the community metabolic simulations. However, these kinds of slow-growing species have critical metabolic contributions to the overall community (e.g. acetogens). It would be helpful to provide information on what species were removed from the simulations and how the simulations changed before and after removing the slow-growing community members.

After checking again, we found that we did not remove any species from the communities, as the growth of every species metabolic model exceeded the 0.001 threshold that we used. We modified our statement accordingly and added the growth rates of the individual species as Supplementary Table S5.

1.11. The presentation of taxa-based results is not informative. Based on the information provided in the Results section, if the raw 16S data were available, a reader would still have to perform their own analysis to get useful information, like taxa bar charts and lists of identified taxa. It would be helpful if Figure 1 contained a new panel summarizing the 16S data in a taxa bar chart and if the authors provided a more extensive summary of the findings in the Results section beyond listing numbers of identified taxa.

In this study, we aim to shift the focus from studying the gut microbiome on the species diversity and taxonomic level to the metabolic network modelling level. And to take a deeper look into the association between the predicted functional analysis of the gut microbiome and T2D. That's why we chose to only present the species that do have significant differences between T2D patients and healthy individuals in our cohort in Figure 1.

We also provide the the table of species abundance per gut community, which would enable some further taxonomic analysis for readers. And we rather wish to keep this paper focused on functional prediction.

1.12. When describing the simulation results, the authors just provide numbers of identified fluxes. Since the models are not included in the supplement, it would be helpful to add a supplemental file with a list of all potential metabolite fluxes, which ones were identified in which simulations, and which were trimmed in the analysis pipeline.

*We thank the reviewer, tables of the predicted fluxes can now be found in:
<https://github.com/Sammerrr/Perfood>*

Grammatical Issues

1.13. Lines 59-62: Run-on sentence that requires rewording for clarity

We modified the sentence for more clarity, lines 59 to 61.

1.14. Lines 78-79: Consistency in plurals

i. Example correction: "Genome-scale metabolite models are powerful tools that provide comprehensive representations of entire metabolic networks of organisms. These models..."

Corrected, lines 77 to 78.

Full Revision

1.15. Line 198: "gender" instead of "Gender"

Corrected.

1.16. Line 211: "Although," instead of "although"

It is "Although" as the reviewer suggests in the original submission.

1.17. Line 296: "butyrate" instead of "Butyrate"

Corrected, line 337.

1.18. Line 298: "alanine" instead of "Alanine"

Corrected, line 341.

1.19. Lines 322-326: Font and text size inconsistent with rest of manuscript

This was a technical error caused by exporting to pdf, we will make sure that this will be corrected.

1.20. Lines 342-366: Font and text size inconsistent with rest of manuscript

Corrected.

Reviewer #1 (Significance (Required))

This paper serves as an intriguing and novel exploration of ecological interactions between host, diet, and microbiomes. Most notably, the integration of human microbiome 16S data to inform community models adds strength to this study. In addition, the literature has previously reported lack of significant difference in diversity measurements between T2D and healthy microbiomes, and this study begins to look at functional differences in this disease that go beyond diversity measures associated with more clinically-focused microbiome studies. The in silico approach to modeling diet-to-microbiome and microbiome-to-host interactions is a nuanced and novel approach. Although there are some concerns with reproducibility of the study given lack of details on EcoGS, lack of availability of raw data, and certain metrics that were calculated using proprietary approaches, this paper adds novel contributions to modeling host-microbiome interactions and elucidating biological mechanisms that drive functional changes in

microbiomes. The audience for this manuscript would be somewhat specialized for those who have experience with in silico metabolic modeling and studying microbial communities. Those involved in translational research might be particularly interested in the work, since a large human cohort is utilized, and potential biomarkers of microbiome dysfunction are identified in the context of a human metabolic disorder. Translational investigators might also use some of the identified pathways and metabolites to alleviate GI sequelae associated with T2D. Those with specific interest in the modeling of microbial communities would also find use in the methods presented as an approach to model host-microbiome interactions. I am a systems biologist, and my field of expertise involves metabolic networking modeling of microbial communities, with an emphasis in gut microbial communities. Although I have limited expertise in the field of T2D specifically, I am well versed in the study of the gut microbiome and in the ways that these communities have been modeled in silico.

We thank the reviewer for this positive assessment of our work.

2. Reviewer #2

Evidence, reproducibility and clarity (Required):

In this study, the authors aim to describe the difference of healthy and diabetic gut microbiota in terms of metabolic modeling. They quantify and compare ecological interactions of microbes in different hosts. Moreover, they use estimated metabolic fluxes to predict the blood glucose level of patients.

Although the authors compiled an impressive amount of raw data, their methodology is poorly explained and thus, in my opinion, their conclusions are not supported well enough. I am especially skeptical about the "precision diet" estimated by the authors. Not only is the method section way too under-detailed to be ever replicated, but also the results of the diet fluxes are not reported. As community models depend on the values of these dietary fluxes, not reporting them is a critical oversight.

Moreover, I am not confident on the baseline blood glucose levels estimation and manual curation, which is not explained and only a non-peer reviewed patent is cited. Here also a more detailed methodology section would increase the quality of the study significantly.

Finally, no code of the analysis (GitHub, ...) and no raw data has been published alongside this paper, which is a must for theoretical analyses.

Major Points

Full Revision

2.1.

Line 114f: As this glucose estimation pipeline is not peer-reviewed, in my opinion, it is fair to include a detailed explanation in here.

* To what extent was the manual curation performed?

We thank the reviewer for the comment, we now better explain how the fasting blood glucose (baseline before meal) and HbA1c levels were estimated, lines 114 to 121:

“Baseline blood glucose levels (before meals) were estimated using a pipeline developed in-house by Perfood, patented and published (20). This machine learning pipeline was trained using tens of thousands of baseline glucose values set by medical diabetologists, as blood glucose values before a meal vary and heavily relies on the correctness of the start time of the meal. In addition, HbA1c levels were collected using continuous glucose monitoring (CGM) sensors and averaged for the aim of this study. Microbiome and metadata for the human cohort can be obtained upon request to Perfood GmbH.”

2.2.

Line 128f, Precision Diet: In my opinion this paragraph is way too under-detailed. Methods should be repeatable. This is not the case here.

We agree with the reviewer, we have rewritten most of the paragraph and renamed it as (Diet estimation, methods, line 132). We removed the precision claim and explained better the steps of estimating the diet and integrating it into the community models. The corresponding code is also made public on github.

2.3.

* What is actually measured in the Perfood digital diary platform?

We removed the mention of the Perfood digital diary platform as we did not use it for creating the diet, we used a reference diet that we share in supplementary table 2 and only the amounts of proteins, carbohydrates, fat and caloric intake were used from the surveyed diet per participant by perfood.

2.4.* What does this in-house dietary ingredient database comprise? Please publish the database in the supplementary material.

We thank the reviewer for this suggestion. We have now clarified how we have derived the estimated diet, The underlying ingredient database is part of the FoCus cohort which is available from the PopGen biobank via a structured application procedure. We have accordingly added the following statement to the manuscript (l. 138 - 140):

“The underlying ingredient database can be obtained through the FoCus cohort (PopGen Biobank, Schleswig-Holstein, Germany) and can be accessed via a structured application procedure (<https://www.uksh.de/p2n/Information+for+Researchers.html>).”

2.5.

* What does the "reference precision diet" comprise out of? Why is it "precise"?

As mentioned in our response to comment 2.2. we have rewritten most of the paragraph and removed the precision claim.

2.6

* What does frequency mean in this context? If it is not per time unit, maybe another word is more suitable.

We changed the term “frequencies” to “amounts”, lines 133, 147, 148 and 150

2.7.

* What is and how big are FOCUS and Perfood cohorts?

We describe the composition of the Perfood cohort in lines 105 to 110 and in supplementary Figure S1 and we added the size of the FOCUS cohort in line 141

2.8.

* Why is averaging performed?

The averaging of the reference diet was performed in order to remove the individual bias as mentioned in line 146.

Considering all the steps necessary to get the estimate, I suggest completely removing the "precision" claim, as I cannot see how this is true.

I find the term "precision" especially hard to believe as you cannot even report units to your fluxes.

Full Revision

As mentioned in our response 2.2, we agree with the reviewer, we have rewritten most of the paragraph and removed the precision claim.

2.9.Line 161f: Here I am missing many points that are explained in Section 3.3. Are they the same? If yes, it may be helpful to refer to them.

We thank the reviewer for this question and all following questions on our EcoGS software.

As mentioned in our response to comment 1.3.

We added the technical details to the description in the GitHub repository (<https://github.com/maringos/EcoGS>):

2.10.

* What is the concrete objective function?

Please refer to our response to comment 1.3

2.11.* How are different microbial abundances handled?

Please refer to our response to comment 1.3

* Are the microbes in two different compartments?

Please refer to our response to comment 1.3

* How do you deal with non-unique solutions?

Please refer to our response to comment 1.3 (Although there are non-unique solutions to such optimization problems, the sybil software in R and the CPLEX solver return stable results. Therefore, in practice, we did not face this issue).

2.14.

Line 233:

* Is this an increase due to an increase in blood glucose?

In our study we can only report on associations but we cannot assume causation, so we cannot be certain whether changes in blood glucose are causal for the changes in microbiome interactions that we observe.

2.15

* Why should there be different levels of blood glucose in healthy subjects?

There are a multitude of factors affecting baseline blood glucose levels including phenotypic characteristics (e.g. obesity), lifestyle and dietary habits (see <https://doi.org/10.1093/ajcn/87.1.258S> for a review). Observing different levels in a heterogenous human cohort is therefore to be expected.

2.16.

* If "healthy" subject have too high glucose baselines are they really healthy or just not diagnosed?

According to established criteria, a fasting blood glucose level above 125 mg/dL is considered as type 2 diabetic. Only two individuals were not diagnosed as diabetic but had blood glucose above 125 mg/dL. Those individuals were removed from the analysis.

Section 4.3:

2.17.

* For met it is not clear if you talk about all models or just a single one.

Corrected

2.18. Did it happen that for certain samples not all exchange reactions you were interested where present? How did you deal with that?

We thank the reviewer for pointing this out, we added the missing information line 303.

Section 4.5:

2.19.

* Line 287: Please add where I can see the H₂S production.

We added this to line 340

2.20.

* Please at p-values when you report significance

Full Revision

We added the P Values in the text when significance is reported.

2.21.

Line 469:

* Are there sulfate-reducing bacteria in your dataset?

Yes, we found two genera in our dataset that are known to reduce sulfate (Desulfovibrio and Bilophila).

2.22.

General:

* please report the significance/ p-value of your regression parameters

We added the P Values in the text when significance is reported.

2.23.

* For me it is not clear, how the linear models are constructed. What is your independent/ what is your dependent variable? Why is this not part of the methods section? Please explain in more detail.

We added the linear model formulas to the methods, section 3.4 lines 229 to 238

2.24

Line 437:

* As high glucose is the main symptom of diabetes, I do not understand how a correlation between fluxes predicting high glucose and being high in diabetics proves any concept.

We thank the reviewer for spotting this. Indeed, this is circular reasoning and we removed the corresponding statement.

Minor Points

2.25.

Line 84: IBD abbreviation not defined

We thank the reviewer for spotting this and have expanded the term to "inflammatory bowel disease".

2.26.

Line 103: As the Perfood company is so central to this study, please give a small overview of Perfood what they do and what "Perfood product" and "Perfood digital diary platform" is supposed to mean.

We added a small overview of Perfood in lines 591 to 595.

2.27.

Line 167: I do not understand the mathematical operations performed, maybe an Equation is warranted.

We added an equation in lines 188 to 193

2.28.

Line 158: Please cite the "logarithmic transformation" study.

We added a citation in line 196

2.29.

Line 176: what is meant by coupling constraints?

We added the citation describing the coupling constraints that we used in line 204

2.30.

Line 187: Filtered out of what?

We changed the statement in line 215 to: "Thereafter, metabolites and reactions were removed if their fluxes had a low standard deviation (<0.01) among the different communities or if they were only present in less than 70% of the diabetic microbial communities".

2.31.

Line 197: Please cite the FDR correction used.

We added the citation to line 225.

2.32.

Line 198 (and others): Is the confounder gender or sex?

We thank the reviewer for pointing this out. We unified the term into “gender”, as participants reported their gender and sex interchangeably because this study does not differentiate between sexes on a chromosomal level.

2.33.

Fig 1, Tab S3:

* Indicate P-value levels (*, **, ***) as common in scientific publications.

We have added the corresponding p-values as Supplementary Table S5 and in <https://github.com/Sammerrr/Perfood>. We prefer not to show those p-values as part of the bar plots with the rather coarse-grained asterisk notation since this is quite uncommon for bar plots.

2.34.

* It would be nice that the significant p-values are part of the text (Line 213) and fig 1 ACD.

The statistical tests and p-values originally reported in Supplementary Table S3. Now we also added them to the text, section 4.1, lines 250 to 255.

2.35.

* In Fig 1A, D you write fold difference, but in the caption you write log₂(fold difference) please clarify.

Corrected.

2.36.

* The categorical coloring of Fig 1A does not work well with the continuous data presented. Please reconsider.

We thank the reviewer for the comment. We agree that the data presented in Figure 1A (the fold difference of bacterial abundance between healthy and diabetic individuals) does not necessarily require categorical colouring. However, we believe that the colouring makes it easier for the reader to visualise which bacteria are increased in diabetic individuals and which ones are increased in healthy individuals.

2.37.

* Fig 1B, as the frequencies are so different in size, please use a log scale on your Y-axis.

We changed the scale to log10 as the reviewer suggested

2.38.

* Are the interaction ratios of CD also log-scale as reported in the methods? If yes, please indicate!

The x axis in figure 1.D is in log2 scale, we added that to the label. The scale in Figure 1.C is not a log scale.

2.39.

Fig 2:

* Please explain what AC, CtH, HtC is. In panel A red is community-to-host, but in Panel B it is "HtC" which may mean host-to-community. What is correct?

Corrected.

2.40.

* Panel B: please write $\log(\text{LM estimate})$ & report either p-values or at least the significance levels. Also the equation is not displayed correctly.

Corrected, the P-values are reported in table S5 and the equation is corrected in line 434

2.41.

* Panel A: Number of *metabolic fluxes* not metabolites?

Corrected in line 422

2.42.

Spelling: CO₂, H₂S, H⁺ all should have the correct subscript.

Corrected.

2.43.

Section 4.4 is missing

Full Revision

Corrected.

Reviewer #2 (Significance (Required)):

I think that the authors have a great dataset at hand. However, the methodology section does not suffice for repeatability of the results.

The authors show that a metabolic disease of the host also affects their microbiome and highlights the importance of microbiome-host interactions during illness.

We thank the reviewer for the valuable comments.

20th Jan 2025

Manuscript Number: MSB-2025-12846-T

Title: Metabolic modelling reveals increased autonomy and antagonism in type 2 diabetic gut microbiota

Dear Dr. Kaleta,

Thank you for the submission of your revised manuscript to Molecular Systems Biology. I am pleased to inform you that we will be able to accept your manuscript pending the following final amendments and appropriate response to reviewers:

1) Please download the EMBO Press "Author Checklist" and complete all relevant questions. This file should be uploaded with your submission. This file can be downloaded from our website at:

<https://www.embopress.org/page/journal/17444292/authorguide>

2) Please upload a .docx or LaTeX file of the main manuscript with no track changes.

3) Please reduce keywords to max. 5.

4) Please include a Data availability section describing how the data, code, model etc. have been made available. This section needs to be formatted according to the example below:

"The datasets and computer code produced in this study are available in the following databases:

- Chip-Seq data: Gene Expression Omnibus GSE46748 (<https://www.ncbi.nlm.nih.gov/geo/query/acc.cgi?acc=GSE46748>)

- Modeling computer scripts: GitHub (<https://github.com/SysBioChalmers/GECKO/releases/tag/v1.0>)

- [data type]: [full name of the resource] [accession number/identifier] ([doi or URL or identifiers.org/DATABASE:ACCESSION])"

5) Please rename the heading "Conflict of Interests" to "Disclosure and competing interests statement". We updated our journal's competing interests policy in January 2022 and request authors to consider both actual and perceived competing interests. Please review the policy <https://www.embopress.org/competing-interests> and update your competing interests if necessary.

6) Author contributions: Please remove it from the manuscript and specify author contributions in our submission system. CRediT has replaced the traditional author contributions section because it offers a systematic machine-readable author contributions format that allows for more effective research assessment. You are encouraged to use the free text boxes beneath each contributing author's name to add specific details on the author's contribution. More information is available in our guide to authors:

<https://www.embopress.org/page/journal/17574684/authorguide#authorshipguidelines>

7) References: Please correct the reference citation in the reference list to be alphabetical (not numerical). Where there are more than 10 authors on a paper, only the first 10 should be listed, followed by "et al.". Please check "Author Guidelines" for more information.

<https://www.embopress.org/page/journal/17574684/authorguide#referencesformat>

8) Our journal encourages inclusion of *data citations in the reference list* to directly cite datasets that were re-used and obtained from public databases. Data citations in the article text are distinct from normal bibliographical citations and should directly link to the database records from which the data can be accessed. In the main text, data citations are formatted as follows: "Data ref: Smith et al, 2001" or "Data ref: NCBI Sequence Read Archive PRJNA342805, 2017". In the Reference list, data citations must be labeled with "[DATASET]". A data reference must provide the database name, accession number/identifiers and a resolvable link to the landing page from which the data can be accessed at the end of the reference. Further instructions are available at .

9) In the Methods, please take care of the following:

- Studies with human research participants: The use of human samples requires information on the authority granting ethics approval (e.g. IRB) and informed consent. If the need for approval is waived, please cite the reason (e.g. non-human subject research because the samples used were de-identified/coded with no identifying information) and legislation in the relevant methods section.

- If the study does indeed include human research participants with informed consent, please also state that the experiments conformed to the principles set out in the WMA Declaration of Helsinki and the Department of Health and Human Services Belmont Report. Please note that this is a separate statement from the specific ethics committee approval and informed consent.

- Please ensure that a statement on whether or not blinding was done is included in the Methods even if no blinding was done. Please also be sure to update the Author Checklist with this information and where it can be found in the manuscript.

10) All Materials and Methods need to be described in the main text using our 'Structured Methods' format. According to this format, the Methods section includes a Reagents and Tools Table (listing key reagents, experimental models, software and relevant equipment and including their sources and relevant identifiers) followed by a Methods and Protocols section describing the methods, ideally using a step-by-step protocol format. The aim is to facilitate adoption of the methodologies across labs. Please download and fill our Reagents and Tools Table template (.docx), which you can find in our author guidelines:

<https://www.embopress.org/doi/10.15252/msb.20178071>. "

11) The section order needs to be corrected. Please place individual sections of the manuscript in the following order: Title page - Abstract & Keywords - Introduction - Results - Discussion - Methods - Data Availability - Acknowledgements - Disclosure and Competing Interests Statement - References - Figure Legends - Expanded View Figure Legends.

12) For the figures and figure legends, please take care of the following:

- Please remove all figures from main manuscript file and leave only main figure legends placed after the references. There are 5 supplementary figures that can be made into Expanded View figures - these should still fit onto one page and be renamed Figure EV1, etc. In that case, the legends should stay in the manuscript, with the heading Expanded View Figures Legends, and placed after the main figure legends. In this case, please be sure to update the callouts of the figures in the manuscript.

Alternatively you may include them in the single Appendix file that also contains the Supplementary Tables. In this case, the nomenclature should be Appendix Figure S1 and Appendix Table S1, etc. The appendix should be uploaded in PDF format and needs to include the title "Appendix for (manuscript title)" along with a table of contents with page numbers.

- Please make sure that all figure callouts are in sequential order.

- Please note that the box plots need to be defined in terms of minima, maxima, centre, bounds of box and whiskers, and percentile in the legends of figures 1B, 2D.

- Please note that information related to n is missing in the legends of figures 1B, 2D, supplementary figure 2.

13) Funding: Please ensure that all funding sources in the manuscript submission system are included in the main manuscript (i.e. please add Christian-Albrechts-Universität zu Kiel (CAU))

14) Synopsis:

- Synopsis image: Please provide a graphic that summarises the main findings of the manuscript on a glance and upload it as a high-resolution jpeg file 550 pixels wide x (300-600) pixels high.

- Synopsis text: Please provide a short standfirst (maximum of 300 characters, including space), limit the bullet points to max. 5 and upload it as a separate .doc file. Please write the bullet points to summarise the key NEW findings. They should be designed to be complementary to the abstract - i.e. not repeat the same text. We encourage inclusion of key acronyms and quantitative information (maximum of 30 words / bullet point). Please use the passive voice.

15) Source Data: Please ensure that a completed Source Data checklist is uploaded as a Related Manuscript File (you will be contacted by my colleague Hannah Sonntag who will send you the checklist). Source Data should be organized as a single source data file (zipped) per figure for main figures (all EV and/or Appendix figure Source Data can be included in a single folder), with the panels clearly visible in the folder structure instead of a single excel file for all Source Data. e.g. all the Source data files for figure 1 need to be saved in a single folder and this needs to be zipped and then uploaded as "SD figure 1.zip" file.

16) As part of the EMBO Publications transparent editorial process initiative (see our policy here:

https://www.embopress.org/transparent-process#Review_Process), Molecular Systems Biology will publish online a Peer Review File (PRF) to accompany accepted manuscripts. This file will be published in conjunction with your paper and will include the anonymous referee reports, your point-by-point response and all pertinent correspondence relating to the manuscript. Let us know whether you agree with the publication of the PRF and as here, if you want to remove or not any figures from it prior to publication. Please note that the Authors checklist will be published at the end of the PRF.

17) Please provide a point-by-point letter INCLUDING my comments as well as the reviewer's reports and your detailed responses (as Word file).

I look forward to reading a new revised version of your manuscript as soon as possible.

Yours sincerely,

Poonam Bheda, PhD
Scientific Editor
Molecular Systems Biology

Reviewer #1:

The authors have adequately addressed comments from both Reviewer 1 and Reviewer 2. However, the authors have an incomplete edit in Lines 241-254, as it seems revisions were made within the previous sentence but the previous sentence components were not removed. Although this caused some confusion in re-reading, it appears to be an easily corrected error.

Reviewer #2:

The authors addressed my comments mostly to my satisfaction.
Some minor points remain:

- * line 207: please remove "precision"
- * line 384: for clarity, a REAME should be added to the GitHub repository, where the different files are explained.
- * line 148: fluxes are usually presented as per hour, thus I was surprised this is not the case here. Can you explain your choice?
- * line 153: please add a reference for the gut microbiome weight.
- * line 196-198: parts of sentences missing.
- * <https://github.com/maringos/EcoGS>: "0: no growth change, 1: increase of growth, 2: decrease of growth" I think, "2" needs to be replaced with "-1" to correspond to the table.

Rev_Com_number: RC-2024-02566

New_manu_number: MSB-2025-12846-T

Corr_author: Kaleta

Title: Metabolic modelling reveals increased autonomy and antagonism in type 2 diabetic gut microbiota

Dear Dr. Bheda,

We thank you very much for the positive assessment of our manuscript. Please find below the responses to your requests.

Please let us know if you need further information.

Best regards,
Christoph Kaleta

1) Please download the EMBO Press "Author Checklist" and complete all relevant questions. This file should be uploaded with your submission. This file can be downloaded from our website at: <https://www.embopress.org/page/journal/17444292/authorguide>

Done

2) Please upload a .docx or LaTeX file of the main manuscript with no track changes.

Done

3) Please reduce keywords to max. 5.

Done

4) Please include a Data availability section describing how the data, code, model etc. have been made available. This section needs to be formatted according to the example below: "The datasets and computer code produced in this study are available in the following databases:

- Chip-Seq data: Gene Expression Omnibus GSE46748

(<https://www.ncbi.nlm.nih.gov/geo/query/acc.cgi?acc=GSE46748>)

- Modeling computer scripts: GitHub

(<https://github.com/SysBioChalmers/GECKO/releases/tag/v1.0>)

- [data type]: [full name of the resource] [accession number/identifier] ([doi or URL or identifiers.org/DATABASE:ACCESSION)]"

Done

5) Please rename the heading "Conflict of Interests" to "Disclosure and competing interests statement". We updated our journal's competing interests policy in January 2022 and request authors to consider both actual and perceived competing interests. Please review the policy <https://www.embopress.org/competing-interests> and update your competing interests if necessary.

Done

6) Author contributions: Please remove it from the manuscript and specify author contributions in our submission system. CRediT has replaced the traditional author contributions section because it offers a systematic machine-readable author contributions format that allows for more effective research assessment. You are encouraged to use the free text boxes beneath each contributing author's name to add specific details on the author's contribution. More information is available in our guide to authors:

<https://www.embopress.org/page/journal/17574684/authorguide#authorshipguidelines>

Done

7) References: Please correct the reference citation in the reference list to be alphabetical (not numerical). Where there are more than 10 authors on a paper, only the first 10 should be listed, followed by "et al.". Please check "Author Guidelines" for more information.

<https://www.embopress.org/page/journal/17574684/authorguide#referencesformat>

Done

8) Our journal encourages inclusion of *data citations in the reference list* to directly cite datasets that were re-used and obtained from public databases. Data citations in the article text are distinct from normal bibliographical citations and should directly link to the database records from which the data can be accessed. In the main text, data citations are formatted as follows: "Data ref: Smith et al, 2001" or "Data ref: NCBI Sequence Read Archive PRJNA342805, 2017". In the Reference list, data citations must be labeled with "[DATASET]". A data reference must provide the database name, accession number/identifiers and a resolvable link to the landing page from which the data can be accessed at the end of the reference. Further instructions are available at

<https://www.embopress.org/page/journal/17574684/authorguide#referencesformat>.

Does not apply

9) In the Methods, please take care of the following:

- Studies with human research participants: The use of human samples requires information on the authority granting ethics approval (e.g. IRB) and informed consent. If the need for approval is waived, please cite the reason (e.g. non-human subject research because the samples used were de-identified/coded with no identifying information) and legislation in the relevant methods section.

- If the study does indeed include human research participants with informed consent, please also state that the experiments conformed to the principles set out in the WMA Declaration of Helsinki and the Department of Health and Human Services Belmont Report. Please note that this is a separate statement from the specific ethics committee approval and informed consent.

- Please ensure that a statement on whether or not blinding was done is included in the Methods even if no blinding was done. Please also be sure to update the Author Checklist with this information and where it can be found in the manuscript.

Our manuscript contains the following statement which we now have moved to the corresponding methods section: "The 16S rRNA sequencing data was generated in the context of a prior project (University of Lübeck, ethics notification ID 20-415). All participants gave their consent to the general use of their anonymised data for scientific purposes. Sequencing data and metadata were provided completely anonymised and thus did not require separate ethical approval according to national regulations for the analysis presented in this manuscript."

10) All Materials and Methods need to be described in the main text using our 'Structured Methods' format. According to this format, the Methods section includes a Reagents and Tools Table (listing key reagents, experimental models, software and relevant equipment and including their sources and relevant identifiers) followed by a Methods and Protocols

section describing the methods, ideally using a step-by-step protocol format. The aim is to facilitate adoption of the methodologies across labs.

Please download and fill our Reagents and Tools Table template (.docx), which you can find in our author guidelines:

<https://www.embopress.org/doi/10.15252/msb.20178071>. "

Done

11) The section order needs to be corrected. Please place individual sections of the manuscript in the following order: Title page - Abstract & Keywords - Introduction - Results - Discussion - Methods - Data Availability - Acknowledgements - Disclosure and Competing Interests Statement - References - Figure Legends - Expanded View Figure Legends.

Done

12) For the figures and figure legends, please take care of the following:

- Please remove all figures from main manuscript file and leave only main figure legends placed after the references. There are 5 supplementary figures that can be made into Expanded View figures - these should still fit onto one page and be renamed Figure EV1, etc. In that case, the legends should stay in the manuscript, with the heading Expanded View Figures Legends, and placed after the main figure legends. In this case, please be sure to update the callouts of the figures in the manuscript. Alternatively you may include them in the single Appendix file that also contains the Supplementary Tables. In this case, the nomenclature should be Appendix Figure S1 and Appendix Table S1, etc. The appendix should be uploaded in PDF format and needs to include the title "Appendix for (manuscript title)" along with a table of contents with page numbers.
- Please make sure that all figure callouts are in sequential order.
- Please note that the box plots need to be defined in terms of minima, maxima, centre, bounds of box and whiskers, and percentile in the legends of figures 1B, 2D.
- Please note that information related to n is missing in the legends of figures 1B, 2D, supplementary figure 2.

Done

13) Funding: Please ensure that all funding sources in the manuscript submission system are included in the main manuscript (i.e. please add Christian-Albrechts-Universität zu Kiel (CAU))

We erroneously mentioned Christian-Albrechts-University as a funder. Since this refers to intramural funds, it can be removed and does not need to be mentioned in the main manuscript.

14) Synopsis:

- Synopsis image: Please provide a graphic that summarises the main findings of the manuscript on a glance and upload it as a high-resolution jpeg file 550 pixels wide x (300-600) pixels high.

- Synopsis text: Please provide a short standfirst (maximum of 300 characters, including space), limit the bullet points to max. 5 and upload it as a separate .doc file. Please write the bullet points to summarise the key NEW findings. They should be designed to be complementary to the abstract - i.e. not repeat the same text. We encourage inclusion of key acronyms and quantitative information (maximum of 30 words / bullet point). Please use the passive voice.

Done

15) Source Data: Please ensure that a completed Source Data checklist is uploaded as a Related Manuscript File (you will be contacted by my colleague Hannah Sonntag who will send you the checklist). Source Data should be organized as a single source data file (zipped) per figure for main figures (all EV and/or Appendix figure Source Data can be included in a single folder), with the panels clearly visible in the folder structure instead of a single excel file for all Source Data. e.g. all the Source data files for figure 1 need to be saved in a single folder and this needs to be zipped and then uploaded as "SD figure 1.zip" file.

Done

16) As part of the EMBO Publications transparent editorial process initiative (see our policy here: https://www.embopress.org/transparent-process#Review_Process), Molecular Systems Biology will publish online a Peer Review File (PRF) to accompany accepted manuscripts. This file will be published in conjunction with your paper and will include the anonymous referee reports, your point-by-point response and all pertinent correspondence relating to the manuscript. Let us know whether you agree with the publication of the PRF and as here, if you want to remove or not any figures from it prior to publication. Please note that the Authors checklist will be published at the end of the PRF.

We agree with the publication of the PRF.

17) Please provide a point-by-point letter INCLUDING my comments as well as the reviewer's reports and your detailed responses (as Word file).

Please see below.

Reviewer #1:

The authors have adequately addressed comments from both Reviewer 1 and Reviewer 2. However, the authors have an incomplete edit in Lines 241-254, as it seems revisions were made within the previous sentence but the previous sentence components were not removed. Although this caused some confusion in re-reading, it appears to be an easily corrected error.

Corrected.

Reviewer #2:

The authors addressed my comments mostly to my satisfaction. Some minor points remain:

* line 207: please remove "precision"

Done

* line 384: for clarity, a REAME should be added to the GitHub repository, where the different files are explained.

Done, the README can now be found on the main page of the repository.

* line 148: fluxes are usually presented as per hour, thus I was surprised this is not the case here. Can you explain your choice?

The reviewer is correct that formal fluxes in metabolic models have the unit mmol per gram dry-weight per hour. However, this is not appropriate in our setting since we use an approximated daily intake of participants as a reference. Thus, our fluxes actually correspond to daily values. We have corrected this accordingly in the methods section.

* line 153: please add a reference for the gut microbiome weight.

We added the following reference:

Sender R, Fuchs S, Milo R. Revised estimates for the number of human and bacteria cells in the body. PLoS Biol. 2016 Aug 19;14(8):e1002533. doi: 10.1371/journal.pbio.1002533. PMID: 27541692

* line 196-198: parts of sentences missing.

Done

* <https://github.com/maringos/EcoGS>: "0: no growth change, 1: increase of growth, 2: decrease of growth" I think, "2" needs to be replaced with "-1" to correspond to the table.

Thank you for spotting this error - this was corrected.

Rev_Com_number: RC-2024-02566

New_manu_number: MSB-2025-12846-T

Corr_author: Kaleta

Title: Metabolic modelling reveals increased autonomy and antagonism in type 2 diabetic gut microbiota

In the Data Availability statement, you mention that the 16S sequences and clinical data are available upon request from perfood GmbH. In general we require sequencing data to be available in a public repository. If the data are subject to controlled access, the Data Availability statement should include the following information: reasons for controlled access, precise conditions of access (including contact details for access requests), a timeframe for response to requests and details of any restrictions imposed on data use via data use agreements.

It would also be helpful to include the same information regarding access to the FoCus cohort information.

We modified the data availability paragraph and added the required informations follows:

- *16S sequences and clinical data, upon request from perfood GmbH (correspondence to Oliver Witt: oliver.witt@perfood.de). Upon request, the data can be obtained within 60 days.*
- *Samples and data for the FoCus cohort were provided by the PopGen Biobank (Schleswig-Holstein, Germany) and can be accessed via a structured application procedure (<https://www.uksh.de/p2n/Information+for+Researchers.html>). Upon application, the FoCus data can be obtained within 7 to 30 days.*
- *Computer scripts, flux exchange predictiondata and statistical results: GitHub (<https://github.com/Sammerrr/Perfood>)*

We did not add a specific reason why the data cannot be shared since public sharing of human clinical data is typically not allowed due to privacy and regulatory standards.

In the section on informed consent, you note that the sequencing data were generated in the context of a prior project. Could you please clarify whether this prior project has been published, and if so, provide a reference to the publication?

We added the citation after the consent statement.

“The 16S rRNA sequencing data was generated in the context of a prior project (University of Lübeck, ethics notification ID 20-415) (Kordowski et al, 2022)”

which is the same reference that we cited in describing the sequence data (Faecal samples were obtained from each participant and processed as described by Kordowski and others (Kordowski et al, 2022))

As previously requested, please ensure that a statement on whether or not blinding was done is included in the Methods even if no blinding was done. Please also be sure to update the Author Checklist with this information and where it can be found in the manuscript.

We added the following statement regarding blinding at the end of the method section, 5.4. Statistical analysis:

“No blinding was necessary in this study, as the data were initially collected from the participants in the context of a service product and were pseudonymised before the analysis.” We also updated the Author Checklist accordingly.

For the synopsis text, please provide a short summary sentence before the bullet points (maximum of 300 characters, including spaces). Please use the passive voice.

Done

Please provide a response to our requests above, indicating how the manuscript has been revised according to our requests.

31st Mar 2025

Manuscript number: MSB-2025-12846R

Title: Metabolic modelling reveals increased autonomy and antagonism in type 2 diabetic gut microbiota

Dear Dr. Kaleta,

Thank you again for sending us your revised manuscript. We are now satisfied with the modifications made and I am pleased to inform you that your paper has been accepted for publication.

Yours sincerely,

Poonam Bheda, PhD
Scientific Editor
Molecular Systems Biology

Rev_Com_number: RC-2024-02566
New_manu_number: MSB-2025-12846R
Corr_author: Kaleta
Title: Metabolic modelling reveals increased autonomy and antagonism in type 2 diabetic gut microbiota